# AUTOMATIC PRIOR SELECTION FOR META BAYESIAN OPTIMIZATION WITH A CASE STUDY ON TUNING DEEP NEURAL NETWORK OPTIMIZERS

## ABSTRACT

The performance of deep neural networks can be highly sensitive to the choice of a variety of meta-parameters, such as optimizer parameters and model hyperparameters. Tuning these well, however, often requires extensive and costly experimentation. Bayesian optimization (BO) is a principled approach to solve such expensive hyperparameter tuning problems efficiently. Key to the performance of BO is specifying and refining a distribution over functions, which is used to reason about the optima of the underlying function being optimized. In this work, we consider the scenario where we have data from similar functions that allows us to specify a tighter distribution a priori. Specifically, we focus on the common but potentially costly task of tuning optimizer parameters for training neural networks. Building on the meta BO method from Wang et al. (2018b), we develop practical improvements that (a) boost its performance by leveraging tuning results on multiple tasks without requiring observations for the same meta-parameter points across all tasks, and (b) retain its regret bound for a special case of our method. As a result, we provide a coherent BO solution for iterative optimization of continuous optimizer parameters. To verify our approach in realistic model training setups, we collected a large multi-task hyperparameter tuning dataset by training tens of thousands of configurations of near-state-of-the-art models on popular image and text datasets, as well as a protein sequence dataset. Our results show that on average, our method is able to locate good hyperparameters at least 3 times more efficiently than the best competing methods.

## 1 INTRODUCTION

The careful tuning of a variety of meta-parameters, such as optimizer parameters and model hyperparameters has become a basic necessity for deep learning (Bergstra et al., 2011; Feurer et al., 2015). Such tuning requires extensive experimentation, retraining models repeatedly with different configurations, and can be challenging at realistic budgets because the tuning landscape is typically non-stationary, noisy and ill-behaved. Tuning has become sufficiently costly that finding more efficient and effective tuning procedures has the potential to save a substantial amount of resources or, alternatively, improve the accuracy of the final models at a given budget.

Some hyperparameters might be common across a large number of tuning problems, such as those pertaining to an optimization algorithm. For example, Adam is used across many deep learning applications and has four parameters that require careful tuning (Nado et al., 2021). Thus, if we have access to the performance of different optimizer-specific hyperparameters on different model training tasks, we may be able to transfer the knowledge among those tasks. This kind of meta-level learning is common among practitioners themselves: when faced with a new tuning problem, one might first try reusing hyperparameter settings that worked well on another problem. The underlying assumption is that hyperparameters should perform similarly across tasks. In this work, we aim to formalize this assumption and automate optimizer hyperparameter tuning by leveraging knowledge from previous experiments. Although our experiments consider optimizer parameter tuning as a practically important sub-problem of hyperparameter tuning for deep neural networks, our method applies to any hyperparameters that are common across multiple tasks.

Bayesian optimization (BO) has become a popular methodology for optimizing the hyperparameters of machine learning models (Snoek et al., 2012; Bergstra et al., 2011) and represents the state-of-the-art (Turner et al., 2021). BO involves specifying a probabilistic model over the function to be optimized and using this to reason about the location of the optimum. The optimization proceeds by iteratively updating the model with new data and using the posterior distribution to reason about where to next evaluate, trading off exploration and exploitation. The model is typically specified with only a-priori assumptions of smoothness, for example using a Gaussian process (GP) with a smooth covariance function. Even if the model is well-specified, BO can be slow to converge due to the generality of the prior assumptions. This would seem wasteful for problems that are repeated often or share considerable structure with previous experiments.

One natural option is to cast our problem as *meta Bayesian optimization*, where the goal is to learn to optimize a function by generalizing from past experience with other similar functions. Indeed, several BO methods exist in the literature, but they are unsuitable for our scenario where we envision potentially thousands of related tasks within e.g. the context of a hyperparameter tuning service. Existing meta BO methods either scale cubically in the number of evaluations and tasks (Swersky et al., 2013; Bardenet et al., 2013) (See §4.3 for more details), impose a restrictive set of assumptions on the available data (Wang et al., 2018b; Swersky et al., 2013) to obtain efficient solutions, or make assumptions on the availability of GP parameters (Volpp et al., 2020) or descriptive task-level features (Brazdil et al., 1994; Bardenet et al., 2013; Yogatama & Mann, 2014).

To address these issues, we introduce HyperBO: a meta BO method that builds upon Wang et al. (2018b) with a relatively simple assumption: all the related functions being optimized are samples from the same GP prior distribution over functions. Concretely, HyperBO assumes the functions are conditionally independent given the hyperparameters, mean and covariance function of the GP. Compared to Wang et al. (2018b), HyperBO does not impose any strict conditions on data or model structures, and a special case of HyperBO retains strong regret bounds. From a computational perspective, HyperBO scales linearly in the number of tasks during training, and does not depend on the number of tasks when deployed. By not imposing assumptions about the data collection conditions, it can be used with large offline datasets or a few related optimization trajectories.

To evaluate HyperBO, we collected a large multi-task hyperparameter tuning dataset by training tens of thousands of configurations of near-state-of-the-art models on popular image and text datasets, as well as on a protein sequence dataset. We compare HyperBO to several hyperparameter tuning baselines in the sequential BO setting. Our results show that optimizers that use hyperparameters suggested by our method are able to obtain better performing models requiring at least 3 times fewer function evaluations than other baselines.

Our main contributions are two-fold: (1) a practical meta BO approach that makes minimal assumptions; and (2) a large multi-task hyperparameter tuning dataset that not only benefits our method but also serves as an ideal benchmark to test future multi-task or meta-learning BO methods.[1]

**Related work**    There is a rich literature of innovative methodologies to improve the efficiency of BO given related tasks or additional context. Here we discuss the most closely related work and explain why these don't solve the specific scenario which we envision. Specifically, our goal is a methodology that is scalable enough to share information across thousands of tasks, each with potentially hundreds of observations, such as in the context of a large BO service or library.

In this work we use the term *meta-BO* to refer to the class of BO methods that use data from existing tasks to optimize a new task. Multi-task BO (Swersky et al., 2013; Poloczek et al., 2017; Yogatama & Mann, 2014) and transfer learning BO using contextual GPs (Krause & Ong, 2011; Bardenet et al., 2013; Poloczek et al., 2016) are both meta BO approaches. Some meta BO methods have also been studied for hyperparameter tuning tasks in machine learning (Feurer et al., 2015).

HyperBO assumes all tasks are independent (after conditioning on the GP), whereas both multi-task and contextual BO rely heavily on the assumption that tasks are related. Thus the latter approaches typically scale cubically in both the number of tasks and observations in each task, meaning that they cannot gracefully scale across both without heavy approximations. When assuming that all inputs are equal across tasks, multi-task BO can be sped up using a Kronecker decomposition of the kernel

---

[1]We are working on open-sourcing the code base and dataset. The dataset is collected based on an open-sourced code base (Gilmer et al., 2021).

to a task kernel and an input kernel which can be inverted separately; a similar assumption is made by Wang et al. (2018b). In comparison, HyperBO scales linearly in the number of tasks (see §B).

End-to-end learning (Chen et al., 2017; Volpp et al., 2020) is another popular meta BO approach for hyperparameter tuning that learns a strategy to suggest new query points based on past history of BO. One limitation of such approaches is that the total number of BO iterations must be determined a-priori. Furthermore, by nature of using a highly parameterized model to train the strategy, we lose the interpretability of intermediate steps that GPs and acquisition functions provide.

HyperBO builds upon Wang et al. (2018b) and Kim et al. (2017; 2019). We resolve their issues with optimizing over a continuous space rather than a discrete set and limitations on using the same set of inputs across tasks. Kim et al. (2017; 2019) estimated a multivariate Gaussian that models values for search strategies in robot manipulation tasks, and thus only considered discrete inputs. Wang et al. (2018b) provided regret bounds for Kim et al. (2017; 2019), which was identified as meta BO without the knowledge of the GP prior. For both finite discrete search spaces and continuous ones, Wang et al. (2018b) requires observations on the same set of inputs across tasks, which is an assumption that is not required for HyperBO; HyperBO still inherits the same regret bound as Wang et al. (2018b) for the special case where the same-inputs assumption is satisfied. Similar ideas also appeared in Perrone et al. (2018), which can be viewed as a special case of HyperBO or Wang et al. (2018b).

## 2 PROBLEM FORMULATION

We consider the standard black-box function optimization scenario: given a real-valued function $f$ defined over a compact, hyper-rectangular space $\mathfrak{X} \subset \mathbb{R}^d$ and given observations of similar functions $f_1, \cdots, f_N$, we seek an $x \in \mathfrak{X}$ optimizing $f$. We inherit our problem formulation from Wang et al. (2018b), but we relax impractical assumptions on data availability (we do not require all observations to be made on the same inputs across tasks) and model restrictions.

**Assumptions and the goal.** Concretely, we assume that there exists a Gaussian process $\mathcal{GP}(\mu, k)$ with unknown mean function $\mu : \mathfrak{X} \to \mathbb{R}$ and kernel $k : \mathfrak{X} \times \mathfrak{X} \to \mathbb{R}$. Let $N$ be the number of tasks and let $M_i$ be the number of observations we have for the $i$th task. Conditioned on independent function samples $f_i \sim \mathcal{GP}(\mu, k)$ and inputs $x_j^{(i)} \in \mathfrak{X}, i \in [N], j \in [M_i]$, we observe evaluations $y_j^{(i)} \sim \mathcal{N}(f_i(x_j^{(i)}), \sigma^2)$ perturbed by *i.i.d.* additive Gaussian noise $\mathcal{N}(0, \sigma^2)$. Taken together, the collection of sub-datasets $D_{f_i} = \{(x_j^{(i)}, y_j^{(i)})\}_{j=1}^{M_i}$ define a dataset $D_N = \{D_{f_i}\}_{i=1}^{N}$. Finally, our goal is to maximize a new function independently sampled from the same GP, $f \sim \mathcal{GP}(\mu, k)$; that is, solve $\arg\max_{x \in \mathfrak{X}} f(x)$ given dataset $D_N$ but unknown functions $\mu, k$ and unknown parameter $\sigma^2$.

**An example.** In our optimizer hyperparameter tuning application, a task corresponds to finding the best optimizer hyperparameters to train a given model on a particular dataset,[2] e.g. training a ResNet (He et al., 2016) on ImageNet (Russakovsky et al., 2015). Notice that we do not assume that the mean function $\mu$, kernel $k$ and noise variance $\sigma^2$ are given. This is consistent with the reality of solving real-world black-box optimization problems including hyperparameter tuning. We must learn those unknown functions and parameters from data. However, in practice, searching in functional spaces to find the right mean $\mu$ or kernel $k$ is a daunting task. Hence for practical concerns, a well defined search space for functions is required. More details on this can be found at §3.1.

**Metrics.** For simplicity, throughout this paper, we focus on the setting where the target function $f$ can only be optimized by iteratively choosing where to evaluate, and defer batch evaluation setups to Sec. F. As we run BO on the target function $f$ for $T$ iterations, we accumulate a set of observations $D_f = \{(x_t, y_t)\}_{t=1}^{T}, y_t \sim \mathcal{N}(f(x_t), \sigma^2)$. We evaluate the quality of the optimization using the *simple regret* metric: $R_T = \max_{x \in \mathfrak{X}} f(x) - f(\hat{x})$, where $\hat{x}$ is the final recommendation at the end of the optimization process. There are various ways of setting $\hat{x}$ based on the observations $D_f$; we use the input that achieved the best evaluation: $\hat{x} = x_\tau; \tau = \arg\max_{t \in [T]} y_t$.

**Bayesian viewpoint.** As mentioned above, the observed functions $f_1, \cdots, f_N$ and the evaluation target $f$ are assumed to be independent draws from the same GP. This assumption is consistent with a hierarchical Bayes interpretation (Fig. 1), where all observed functions are independent conditioned on the GP. Notice that for BO, each selected input $x_j^{(i)}$ depends on all previous observations. But we only describe the generative model of a hierarchical GP for simplicity.

---

[2]Technically, we also consider different batch sizes to be different tasks.

More specifically, we assume that underlying functions of hyperparameter optimization tasks are defined by a parameter $\theta \sim p(\theta; \alpha)$; mean and kernel functions $\mu$ and $k$ are defined by deterministic functions parameterized by $\theta$. The independent function samples $\{f_i\}_{i \in [N]}$ are draws from $\mathcal{GP}(\mu, k)$. We then learn function $f$ from observations $D_N$ on all other conditionally *i.i.d.* function samples $f_1, \cdots, f_N$. We forgo a fully Bayesian approach that samples from the posterior over $\theta$ at every BO iteration. Our method can be viewed as a type-II maximum likelihood approximation of such a Bayesian solution.

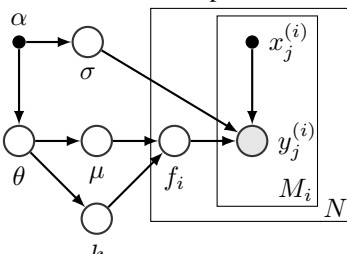

Figure 1: Graphical model for a hierarchical Gaussian process.

**Notations.** Let $[n]$ denote $\{1, \cdots, n\}, \forall n \in \mathbb{Z}^+$. For conciseness, we write the evaluation of a function $f : \mathfrak{X} \to \mathbb{R}$ on matrix $\boldsymbol{x} = [x_i]_{i=1}^n \in \mathbb{R}^{n \times d}$ as $\mu(\boldsymbol{x}) := [\mu(x_i)]_{i=1}^n$. Similarly, for two matrices (i.e., the concatenation of input vectors) $\boldsymbol{x}, \boldsymbol{x}'$, we write the corresponding kernel matrix as $k(\boldsymbol{x}, \boldsymbol{x}') := [k(x_i, x'_j)]_{i \in [n], j \in [n']}$, and shorten $k(\boldsymbol{x}) := k(\boldsymbol{x}, \boldsymbol{x})$. We denote a (multivariate) Gaussian distribution with mean $u$ and variance $\Sigma$ by $\mathcal{N}(u, \Sigma)$, and a Gaussian process (GP) with mean function $\mu$ and covariance function $k$ by $\mathcal{GP}(\mu, k)$. Let $\sigma^2$ be the noise variance in observations. Given a set of observations $D = \{(x_t, y_t)\}_{t=1}^T, \boldsymbol{y}_T = [y_t]_{t=1}^T \sim \mathcal{N}(f(\boldsymbol{x}_T), \sigma^2 \boldsymbol{I}), \boldsymbol{x}_T = [x_t]_{t=1}^T$ and $f \sim \mathcal{GP}(\mu, k)$, we denote the corresponding conditional GP distribution as $\mathcal{GP}(\mu, k \mid D)$. Let $\psi(x) = k(x, \boldsymbol{x}_T)(k(\boldsymbol{x}_T) + \sigma^2 \boldsymbol{I})^{-1}$. Recall that the conditional distribution, $\mathcal{GP}(\mu, k \mid D) = \mathcal{GP}(\mu_D, k_D)$, is given for any $x, x' \in \mathfrak{X}$ as

$$\mu_D(x) = \mu(x) + \psi(x)(\boldsymbol{y}_T - \mu(\boldsymbol{x}_T)), \quad k_D(x, x') = k(x, x') - \psi(x)k(\boldsymbol{x}_T, x'). \tag{1}$$

## 3 OUR METHOD

As shown in Alg. 1, our approach trains the GP hyperparameters on a representative set of datasets and fixes them for the duration of the optimization procedure; we refer to this approach as HyperBO. HyperBO runs in two steps. First, we learn a GP model $\mathcal{GP}(\hat{\mu}, \hat{k})$ to approximate the ground-truth (unknown) GP that generated the dataset $D_N$. Then, we do standard BO to optimize a new function $f$ with the learned GP $\mathcal{GP}(\hat{\mu}, \hat{k})$. The initial learning process (line 2) is the critical difference between HyperBO and standard BO algorithms, as well as the key contribution of this paper.

**Algorithm 1** HyperBO with acquisition function $\alpha(\cdot)$.

1: **function** HYPERBO $(f, D_N)$
2:     $\mathcal{GP}(\hat{\mu}, \hat{k}) \leftarrow$ TRAINGP$(D_N)$
3:     $D_f \leftarrow \emptyset$
4:     **for** $t = 1, \cdots, T$ **do**
5:         $x_t \leftarrow \underset{x \in \mathfrak{X}}{\arg\max} \, \alpha\left(x; \mathcal{GP}(\hat{\mu}, \hat{k} \mid D_f)\right)$
6:         $y_t \leftarrow$ OBSERVE$(f(x_t))$
7:         $D_f \leftarrow D_f \cup \{(x_t, y_t)\}$
8:     **end for**
9:     **return** $D_f$
10: **end function**

Based on the Bayesian graphical model interpretation (Fig. 1), our goal is to obtain a point estimate $\hat{\theta}$ for the parameter $\theta$. Given this estimate, we can then estimate the mean function $\hat{\mu}$ and the kernel $\hat{k}$, which defines our learned model $\mathcal{GP}(\hat{\mu}, \hat{k})$. During the BO iterations (Alg. 1, lines 4-8), we update the conditional GP, but do not re-estimate the GP mean and kernel. By separating the data for the conditional GP update and GP parameter training, we minimize the computational cost while still maintaining good performance both theoretically and empirically. Moreover, we avoid the BO chicken-and-egg dilemma (Wang et al., 2018b) where the search strategy is trained on data collected in the BO process and the data points are selected by the search strategy.

Next, we introduce our GP training strategy based on two types of objectives: marginal data likelihood (§ 3.1) and distance between estimates and model predictions (§ 3.2). In the appendix, §B reveals the complexity of HyperBO that is linear in the number of tasks and § 3.3 shows that a special case of HyperBO retains strong regret bounds (Wang et al., 2018b).

### 3.1 MARGINAL LIKELIHOOD

A straightforward way to train a GP is by optimizing the log marginal likelihood over the GP's hyperparameters. This is also known as type II maximum likelihood approximation (Rasmussen

& Williams, 2006). In our case, we derive the data likelihood for the observations from multiple functions that are assumed to be given, which is a key difference to regular GP or BO setups. The log marginal likelihood for our method is

$$\log p(D_N \mid \mu, k, \sigma^2) = \sum_{i=1}^{N} \left( -\frac{1}{2} \bar{\boldsymbol{y}}_{(i)}^{\top} K^{-1} \bar{\boldsymbol{y}}_{(i)} - \frac{1}{2} \log |K| - \frac{M_i}{2} \log 2\pi \right), \quad (2)$$

where $\bar{\boldsymbol{y}}_{(i)} = \boldsymbol{y}^{(i)} - \mu\left(\boldsymbol{x}^{(i)}\right)$, $K = k\left(\boldsymbol{x}^{(i)}\right) + \sigma^2 \boldsymbol{I}$, $\boldsymbol{x}^{(i)} = [x_j^{(i)}]_{j=1}^{M_i}$ and $\boldsymbol{y}^{(i)} = [y_j^{(i)}]_{j=1}^{M_i}$.

Our solution to the choice of mean function, kernel function and noise variance then becomes

$$\hat{\mu}, \hat{k}, \hat{\sigma}^2 = \arg\max_{\mu, k, \sigma^2} \log p(D_N \mid \mu, k, \sigma^2). \quad (3)$$

For mean function $\mu$ and kernel $k$, this optimization is done in functional space. While methods exist to search for functional structures (Kemp & Tenenbaum, 2008; Malkomes & Garnett, 2018), one may opt for a simple search strategy within a group of functional structures (e.g. mean $\mu \in \{\text{linear}, \text{constant}\}$ and kernel $k \in \{\text{exponentiated quadratic}, \text{Matérn}\}$). For all combinations of mean/kernel structures or functional classes, we then optimize the parameterization of them and noise variance $\sigma^2$ to eventually solve Eq. 3. Details of how we defined the search space can be found in §4.

## 3.2 DISTANCE BETWEEN ESTIMATES AND MODEL PREDICTIONS

Although the marginal likelihood is a straightforward objective to optimize, it may not be straightforward to interpret how high of a likelihood is high enough for us to stop our search for a decent model. Nevertheless, we may be able to directly estimate the sample mean and covariance, and the distance between those estimates and model predictions could be a good indicator of how good the model is. We will show in §3.3 that a distance objective may lead us to better theoretical properties.

Here we consider a special case of dataset $D_N$ where part of it has matching inputs across some sampled functions. More formally, suppose we have a *matching dataset* $D'_N = \{(x_j, \boldsymbol{y}_j)\}_{j=1}^{M}$ where $M$ is a positive integer, $x_j \in \mathfrak{X}$, $\boldsymbol{y}_j = [y_j^{(i)}]_{i=1}^{N} \in \mathbb{R}^N$ and $y_j^{(i)} \sim \mathcal{N}(f(x_j), \sigma^2)$. Empirically, dataset $D'_N$ can be constructed by querying a set of functions $f_1, \cdots, f_N$ at the same set of input locations $\boldsymbol{x} = [x_j]_{j=1}^{M} \in \mathbb{R}^{M \times d}$ to obtain an observation matrix $\boldsymbol{y} = [\boldsymbol{y}_j]_{j=1}^{M} \in \mathbb{R}^{M \times N}$.

By definition of a GP, the vector of all function queries $f(\boldsymbol{x})$ is distributed according to a multivariate Gaussian distribution $\mathcal{N}(\mu(\boldsymbol{x}), k(\boldsymbol{x}))$. With our observation model, we get the distribution for observations $\boldsymbol{y} \sim \mathcal{N}(\mu(\boldsymbol{x}), k(\boldsymbol{x}) + \boldsymbol{I}\sigma^2)$ for some unknown mean function $\mu$ and kernel $k$.

However, given that we have access to all observations $\boldsymbol{y}$, we can estimate the mean on inputs $\boldsymbol{x}$ as $\tilde{\boldsymbol{\mu}} = \frac{1}{N} \boldsymbol{y} 1_N \in \mathbb{R}^M$ and estimated covariance as $\tilde{K} = \frac{1}{N}(\boldsymbol{y} - \tilde{\boldsymbol{\mu}} 1_N^{\top})(\boldsymbol{y} - \tilde{\boldsymbol{\mu}} 1_N^{\top})^{\top} \in \mathbb{R}^{M \times M}$; here $1_N$ is a column vector of size $N$ filled with 1s. We use a biased estimate of covariance to be consistent with the corresponding maximum likelihood estimator in Eq. 3. [3] Notice that the estimated covariance includes in diagonal terms the variance of the observation noise.

For any distance function between the estimate $\mathcal{N}(\tilde{\boldsymbol{\mu}}, \tilde{K})$ and model prediction $\mathcal{N}(\mu(\boldsymbol{x}), k(\boldsymbol{x}) + \boldsymbol{I}\sigma^2)$, we obtain an objective to minimize, $\mathcal{D}\left(\mathcal{N}(\tilde{\boldsymbol{\mu}}, \tilde{K}), \mathcal{N}(\mu(\boldsymbol{x}), k(\boldsymbol{x}) + \boldsymbol{I}\sigma^2)\right)$. While there are different measures of distributional discrepancy, we adopt the KL divergence. Let $\boldsymbol{\mu} = \mu(\boldsymbol{x})$ and $K = k(\boldsymbol{x}) + \boldsymbol{I}\sigma^2$. The KL divergence is defined as

$$\mathcal{D}_{\text{KL}}\left(\mathcal{N}(\tilde{\boldsymbol{\mu}}, \tilde{K}), \mathcal{N}(\boldsymbol{\mu}, K)\right) = \frac{1}{2}\left(\text{tr}(K^{-1}\tilde{K}) + (\boldsymbol{\mu} - \tilde{\boldsymbol{\mu}})^{\top} K^{-1}(\boldsymbol{\mu} - \tilde{\boldsymbol{\mu}}) + \ln \frac{|K|}{|\tilde{K}|} - M\right), \quad (4)$$

and we can estimate the mean, kernel and noise variance by minimizing $\mathcal{D}_{\text{KL}}$. While it is difficult to gauge how much a probability density is enough to obtain a good model, Eq. 4 is a "distance" that goes to 0 as the difference between two distributions reduces. One may choose to do early stopping or model selection based on how close Eq. 4 is to 0. Through information theory, we also know that the KL divergence in Eq. 4 describes the number of extra bits (or nats) to encode the multivariate normal $\mathcal{N}(\tilde{\boldsymbol{\mu}}, \tilde{K})$. Overall we found the KL divergence in Eq. 4 relatively more interpretable than the marginal likelihood in Eq. 3.

---

[3]One may choose to re-scale learned kernel by $\frac{N}{N-1}$ to be unbiased.

The KL divergence in Eq. 4 introduces a different optimization landscape than the marginal likelihood in Eq. 3. The KL divergence also makes use of the matching dataset $D'_N$ in a way that the marginal likelihood cannot. In fact, all matching inputs in the marginal likelihood in Eq. 3 are implicit: all inputs are passed in to mean/kernel functions, and so there is no way that Eq. 3 can be informed that some inputs are the same across tasks. As shown in §4, the KL divergence in Eq. 4 interestingly led to better results in our experiments.

### 3.3 THEORETICAL ANALYSES

While it is nontrivial to prove regret bounds for general scenarios without strict assumptions, it is straightforward to show a regret bound for our method with objective $\mathcal{D}_{\mathrm{KL}}$ of Eq. 4 in the matching dataset case where BO is running on a finite set of inputs (Wang et al., 2018b).

**Theorem 1.** *Let* $N \geq 4 \log \frac{6}{\delta} + T + 2$. *With probability at least* $1 - \delta$, *simple regret in* $T$ *iterations of Alg. 1 with special cases of either GP-UCB or PI satisfies*

$$R_T < O\left(\sqrt{\frac{1}{N-T}} + \sqrt{\log\frac{1}{\delta}}\right) O\left(\frac{1}{2T} \max_{A \subset \mathfrak{X}, |A|=T} \log |\boldsymbol{I} + \sigma^{-2}k(A)| + \sigma\right). \qquad (5)$$

More details can be found at §D. Theorem 1 shows that the regret bound has a linear dependency on the observation noise $\sigma$. This is expected because in practice, we select the best observation rather than best function value (before observing a noisy version of it) to compute the simple regret. Another reason is that we learn the noise parameter $\sigma$ jointly with the kernel, as shown by Eq. 4. Hence when computing acquisition functions, the noise $\sigma$ is always included in the predicted variance.

Intuitively, the more sub-datasets we have in the dataset, the larger $N$ is, the better we are able to estimate the GP model, and the closer the regret bound is to the case where the GP model is assumed known. Interestingly, the number of BO iterations $T$ makes the regret smaller in the second term but larger in the first term in Eq. 5. Usually as we get more observations, we get more information about the maximizer, and we are able to optimize the function better. However, as we get more observations on the new function, GP conditional predictions have more freedom to deviate from the ground truth (see Lemma 1 of Wang et al. (2018b)). As a result, we get less and less confident about our predictions, which is eventually reflected in a looser regret upper bound.

It is tempting to prove similar bounds for more general settings where inputs are not the same across all sub-datasets and BO happens in continuous space. Though the only prerequisite is to show that the difference between the learned mean/kernel and the ground truth mean/kernel is small, this prerequisite is as difficult as showing we can find a model that has bounded generalization error across the entire continuous input space of an arbitrary function. Instead of making unrealistic assumptions just to satisfy such prerequisite, we leave the regret bound for general settings as an open question.

## 4 EXPERIMENTS

Our goal in this paper is to provide a practical approach for hyperparameter optimization when we are given data on a range of tasks over the same search space. To analyze the effectiveness of our proposal, we take the optimizer hyperparameter tuning problem in deep learning as a case study. Our implementation of HyperBO is based on JAX (Bradbury et al., 2018).[4]

To reduce ambiguity, we distinguish between datasets that individual neural networks are trained on and the dataset we collected that includes optimizer hyperparameter points with their validation errors (and other metrics). We will call the former (e.g. MNIST, CIFAR10) task datasets and call the latter the tuning dataset. The tuning dataset is what we described as dataset $D_N$ in §2.

### 4.1 HYPERPARAMETER TUNING DATASET

In order to collect our hyperparameter tuning dataset, the PD1 Neural Net Tuning Dataset, we defined a set of 24 neural network tuning tasks[5] and a single, broad search space for Nesterov

---

[4]We are working on open-sourcing our code as well as trained GP models.

[5]The batch size 1024 ResNet50 ImageNet task only has 100 hyperparameter points because we abandoned it when scaling up data collection in order to save compute resources. It is used in training, but not evaluation.

momentum. Each task is defined by a task dataset (e.g. ImageNet), a specific neural network model (e.g. ResNet50), and a batch size. Tab. 1 shows all the tasks that we consider in the tuning dataset. We used an existing code base (Gilmer et al., 2021) for neural network model training. The dataset used roughly 12,000 machine-days of computation for approximately 50,000 hyperparameter evaluations.

For each task, we trained the model on the task dataset repeatedly using Nesterov momentum (Nesterov, 1983; Sutskever et al., 2013), with the task's minibatch size, with different hyperparameter settings drawn from the 4-dimensional search space detailed in Tab. 2. We tuned the base learning rate, $\eta$, on a log scale, the momentum, $\beta$, with $1 - \beta$ on a log scale, and the polynomial learning rate decay schedule power $p$ and decay steps fraction $\lambda$. We used a polynomial decay schedule with the following form: $\eta_\tau = \frac{\eta}{1000} + \left(\eta - \frac{\eta}{1000}\right)\left(1 - \frac{\min(\tau, \lambda\mathcal{T})}{\lambda\mathcal{T}}\right)^p$, where $\tau$ is the training step and $\mathcal{T}$ is the total number of training steps for the task.

Table 1: Tasks

| Task Dataset | Model | Batch Sizes |
|---|---|---|
| CIFAR10 | Wide ResNet | {256, 2048} |
| CIFAR100 | Wide ResNet | {256, 2048} |
| Fashion MNIST | Max pool CNN ReLU | {256, 2048} |
| Fashion MNIST | Max pool CNN tanh | {256, 2048} |
| Fashion MNIST | Simple CNN | {256, 2048} |
| ImageNet | ResNet50 | {512, 1024, 2048} |
| LM1B | Transformer | {2048} |
| MNIST | Max pool CNN relu | {256, 2048} |
| MNIST | Max pool CNN tanh | {256, 2048} |
| MNIST | Simple CNN | {256, 2048} |
| SVHN (no extra) | Wide ResNet | {256, 1024} |
| WMT15 German-English | xformer | {64} |
| uniref50 | Transformer | {128} |

Table 2: 4-dimensional input search space (see text for more details)

| Hyperparameter | Range | Scaling |
|---|---|---|
| $\eta$ | $[10^{-5}, 10]$ | Log |
| $p$ | $[0.1, 2.0]$ | Linear |
| $1 - \beta$ | $[10^{-3}, 1.0]$ | Log |
| $\lambda$ | $[0.01, 0.99]$ | Linear |

We collected two types of data: matched and unmatched data. Matched data used the same set of uniformly-sampled hyperparameter points across all tasks and unmatched data sampled new points for each task. All other training pipeline hyperparameters were fixed to hand-selected, task-specific default values. All of our tasks are classification problems, so they all used the same training loss, although occasionally task-specific regularization terms were added. For each trial (training run for a single hyperparameter point), we recorded validation error (both cross entropy error and misclassification rate). In many cases, poor optimizer hyperparameter choices can cause training to diverge. We detected divergent training when the training cost became NaN and then marked the trial but did not discard it. Please see the Appendix, supplementary material, and code (Onomous, 2021) for additional details about the tasks and training procedure. The different tuning tasks vary in difficulty and numbers of data points, but generally there are roughly 500 matched datapoints and 1500 unmatched datapoints per tuning task. For unmatched data only, we attempted to generate roughly similar numbers of non-divergent points across tasks, so tasks with a higher probability of sampling a hyperparameter point that causes training to diverge will tend to have more trials.

## 4.2 DESCRIPTION OF ALL COMPARED METHODS

Our method HyperBO has several variants including using different acquisition functions and different objectives. In §4, unless otherwise mentioned, we used a thresholded probability of improvement (PI) as the acquisition function. We set PI in line 5 of Alg. 1 as $\frac{\hat{\mu}_{D_f}(x) - \max_t(y_t + 0.1)}{\hat{\sigma}_{D_f}(x)}$. We empirically evaluated a variety of acquisition functions, but found PI thresholded at 0.1 to be surprisingly effective. Because we model the observations as log error rate, this actually trades off exploration and exploitation - i.e. with larger error rates this seeks relatively more substantial improvements than with small error rates. The list of 5 different acquisition functions we tested is as follows: PI with 0.1 threshold, expected improvement and UCB with 2, 3, 4 coefficients. See their comparisons at §E.5.

We use H* NLL to denote HyperBO with negative log marginal likelihood as the objective and H* KL to denote HyperBO with KL divergence on matching datapoints as objective. Both objectives are optimized via L-BFGS (Nocedal, 1980) with $\mu(x) = \theta_0^\top \tanh(\theta_1^\top x), k(x, x') = \text{Matérn32}(\tanh(\theta_1^\top x), \tanh(\theta_1^\top x')), \theta_1 \in \mathbb{R}^{4 \times 8}$. These two settings of HyperBO are relatively representative of the performance of variants of HyperBO. See comparisons over objectives at §E.4.

Our baselines include (a) Rand: Random search in the corresponding scaled space in Tab. 2. (b) STBO: Single-task BO where in every BO iteration, STBO optimizes the GP hyperparameters via marginal likelihood on data of the test task. This implementation corresponds to the basic off-the-shelf BO setups. (c) STBOH: Single-task GP-UCB with a *hand-tuned* prior on hyper-parameters including

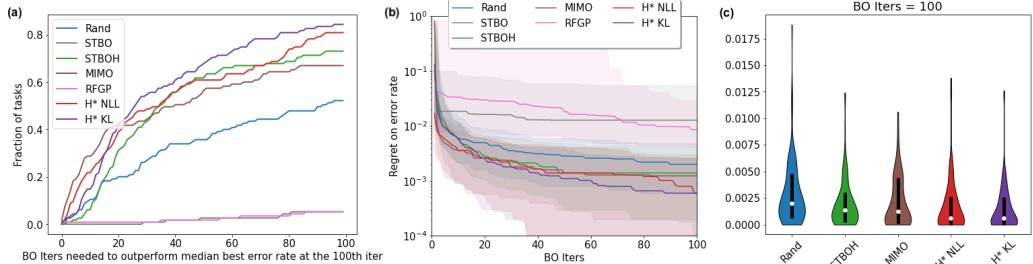

Figure 2: (a) The performance profile for outperforming the median of best error rates at the 100th BO iteration. (b) BO convergence of all methods: the median and 20/80 percentiles of the regrets on error rates over 115 BO runs: 23 tasks and each with 5 repeats of different random seeds. (c) A violin plot on the vertical slices of (b) at the 100th iteration; the white dot is the median and the black line is the 20/80 percentile. Overall, HyperBO methods H* NLL and H* KL are able to achieve the lowest regret on error rate on the majority of tasks with fewer iterations.

UCB coefficient. (d) MIMO: Multi-task BO with GP bases as an ensemble of feedforward neural networks with shared subnetworks (Havasi et al., 2020). (e) RFGP: Multi-task BO with GP bases as random features. More details can be found at §E.1.

## 4.3 RESULTS ON OFFLINE OPTIMIZER HYPERPARAMETER TUNING TASKS

Many tasks in §4.1 can use up a lot of compute resources and time, which makes it infeasible to perform a wide variety of experiments to analyze the characteristics of BO methods. Hence we adopt an offline approximation, which runs BO only on the finite set of points that each tuning sub-dataset contains. More details and analyses are available in Appendix E.2.

**Holding out relevant tasks.** Fig. 2 (a) shows the *performance profiles*, the fraction of all test tasks that each method from §4.2 is able to outperform a baseline criterion at each BO iteration. We can see that MIMO is able to outperform other methods in the beginning 20 BO iterations, but its leading position soon gets surpassed by HyperBO (H* NLL and H* KL). Fig. 2 (b,c) illustrates the BO convergence curves of all competing methods, together with the vertical slice at the 100th iterations. RFGP and STBO are both falling much behind Rand. STBO trains the GP on the data that the GP suggests to query. Optimizing the marginal data likelihood on at most 100 datapoints in fact may not lead to a better model than random initialization (see Tab. 5 in §F). Surprisingly, the contextual information learned by RFGP did not generalize to a new task. On the other hand, MIMO is able to obtain a slightly better error rate than STBOH. Overall, learning the GP prior through data as with HyperBO outperforms other meta BO methods, and is a more principled and effective way to obtain the GP prior when compared with hand-tuning.

**Effect of number of training tasks.** We now investigate the impact of the number of training tasks on the performance of meta BO methods. In Fig 3 we show the BO simple regrets on tasks from Table 1 (except ImageNet ResNet50 2048) that use meta BO models trained on different number of training tasks. To analyze the performance of all methods on less-related tasks, we first remove training tasks that have the same task dataset as our current tuning task for testing, and then remove randomly selected training datasets from the rest.

Figure 3: Medians and 20/80 percentiles of regrets on best validation error rates for methods that uses models trained on 3 to 23 training tasks.

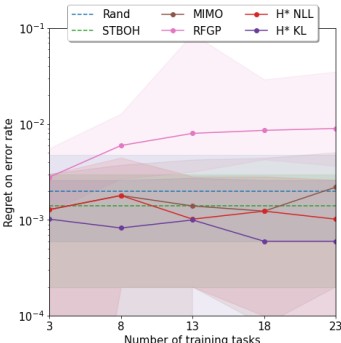

HyperBO variants reduced the simple regret as more training tasks are given. Interestingly, H* NLL and H* KL are already slightly better than Rand and STBOH when they started off with only 3 training tasks. There are reasonable fluctuations in the results but overall the trend of regret is going down as the number of training tasks increases. MIMO also reduced regret when the number of tasks increased from 8 to 18. RFGP, however, fails to learn from training tasks possibly because it did not learn good task embeddings for GP regression models.

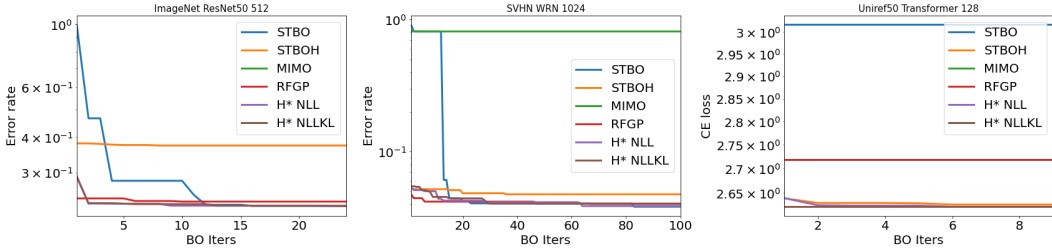

Figure 5: Results of running BO methods in the online setting on 3 different tasks. The image based tasks use best validation error rate as objective while the text based ones including Uniref50 use best validation CE loss. In all 3 tasks, HyperBO methods achieved better results.

**Effect of number of data points in training tasks.** One remaining question is, how does $M_i$ in §2, the number of data points in each training tasks, affect the performance of meta BO methods. We analyze the impact of $M_i$ by removing a portion of all data that we have access to for each task. In particular, we set the percentage of remaining data to be $0.2\%, 0.5\%, 1\%, 3\%, 5\%, 10\%, 30\%, 50\%, 70\%, 90\%$. Remaining datapoints are selected uniformly randomly, which breaks the structure of matching data. Hence we do not include H* KL in this comparison, as H* KL only makes use of matching data.

Fig. 4 shows how the simple regret changes as the fraction of training data grows. Below $10\%$ training data, we observe clear trend that more data lead to lower regret for both H* NLL and MIMO, and relatively no change for RFGP. We also found that the performance of HyperBO (H* NLL) does not change much as the fraction of training data increases from $5\%$ to $90\%$. However, MIMO and RFGP suffers significantly from more data as the fraction of training data increases from $5\%$ to $50\%$. It is not entirely clear why MIMO and RFGP have such behaviors. One conjecture is that neural network based Bayesian linear regression models may get too confident once the amount of data reaches a certain threshold. This means much less exploration if those models are used for BO.

Figure 4: Medians and 20/80 percentiles of simple regrets for methods that uses models trained on $0.2\%$ to $90\%$ of data in each task.

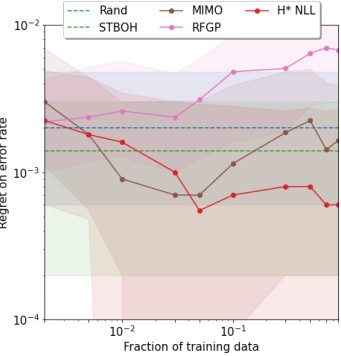

### 4.4 RESULTS ON ONLINE OPTIMIZER HYPERPARAMETER TUNING TASKS

We now evaluate HyperBO methods, H* NLL and H* NLLKL (NLL plus KL divergence on matching datapoints as the objective), in the online setting, where we optimize over the full hypercube and some hyperparameters may be *infeasible* to evaluate. See full results at §E.3. Fig. 5 shows that it can be difficult for STBO or MIMO to recover from a "bad" datapoint, but HyperBO methods are robust and performed the best among all methods being compared.

## 5 DISCUSSION AND CONCLUSION

While we focused on obtaining a better prior in BO in this work, the following directions are orthogonal to what we studied: different search spaces across tasks, batch evaluation, high-dimensional or large scale data, etc. However, it should be straightforward to combine their solutions with HyperBO. Please find more discussions at §F together with implications of our assumptions.

HyperBO is a novel meta BO approach that supports practical applications that involve continuous inputs queried at possibly non-aligned locations across tasks. HyperBO uses a simple yet effective idea that is easy to implement and efficient to run. We evaluated HyperBO on real-world big model optimizer tuning tasks, and the results demonstrated its superior performance over state-of-the-art competing methods.

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

## A  HIERARCHICAL GP, A BAYESIAN PERSPECTIVE

In Bayesian viewpoint of §2, we assume that the overall setting of the hyperparameter optimization task is defined by a parameter $\theta \sim p(\theta; \alpha)$; mean and kernel functions $\mu$ and $k$ are drawn from $p(\mu, k \mid \theta)$. The independent function samples $\{f_i\}_{i \in [N]}$ are themselves draws from $\mathcal{GP}(\mu, k)$. The generative story is as follows:

- Draw GP parameter $\theta$ from $p(\theta; \alpha)$ and observation noise parameter $\sigma$ from $p(\sigma; \alpha)$.
- Draw mean function $\mu$ and kernel function $k$ from $p(\mu, k \mid \theta)$.
- For each task $i$ from 1 to $N$,
    - Draw a function $f_i$ from $\mathcal{GP}(\mu, k)$.
    - For each data point $j$ from 1 to $M_i$,
        * Given input $x_j^{(i)}$, we draw the observation $y_j^{(i)} \sim \mathcal{N}(f_i(x_j^{(i)}), \sigma^2)$.

We simplify this hierarchical setting by defining $p(\mu, k \mid \theta)$ to be a sum of Dirac delta functions: both mean function $\mu$ and kernel $k$ are deterministic functions parameterized by $\theta$. Thus, we can infer GP parameter $\theta$ and noise $\sigma$ from their posterior $p(\theta, \sigma \mid D_N \cup D_f; \alpha)$ and obtain an informed prediction for the target function

$$p(f \mid D_N \cup D_f) = \int_\theta p(f \mid \theta) p(\theta \mid D_N \cup D_f; \alpha)$$
$$= \int_\theta p(f \mid \theta) \int_\sigma p(\theta, \sigma \mid D_N \cup D_f; \alpha)$$

In other words, we learn function $f$ from observations on all other conditionally *i.i.d.* function samples $f_1, \cdots, f_N$. We forgo a fully Bayesian approach that samples from the posterior over $\theta$ at every BO iteration, although our method, HyperBO, can be viewed as a type-II maximum likelihood approximation of such a Bayesian solution.

## B  COMPUTATIONAL COMPLEXITY

The marginal likelihood in Eq. 2 naturally decomposes into a sum of GP data likelihood terms on each sub-dataset $D_{f_i}$. The time complexity to compute Eq. 2 is $\mathcal{O}(M^3 N)$, where $N$ is the number of sub-datasets and $M = \max_{i=1}^N M_i$ is the maximum number of data points for these sub-datasets. Notice that our method scales linearly in the number of tasks, $N$, in contrast to the cubic $\mathcal{O}(M^3 N^3)$ scaling of multi-task or contextual BO methods (Swersky et al., 2013; Bardenet et al., 2013; Poloczek et al., 2016; Yogatama & Mann, 2014). The only cubic cost of HyperBO is on the number of data points in sub-datasets.

To train a GP with $K$ optimization steps on Eq. 2, the time complexity is $\mathcal{O}(M^3 N K)$. The distance regularizers introduced in §3.2 requires estimating mean and covariance, which takes $\mathcal{O}(M^2 N)$ for matrix multiplication. The KL divergence in Eq. 4 has complexity $\mathcal{O}(M^3)$ to compute and $\mathcal{O}(M^3 K)$ to optimize.

If there is any better probabilistic model than a GP to fit the data with less compute time, we can easily swap it in and reduce the $\mathcal{O}(M^3)$ complexity that the GP contributed to the $\mathcal{O}(M^3 N)$ complexity of Eq. 2. For example, if we approximate a GP with a linear model on $V$ random features (Rahimi et al., 2007), the complexity of Eq. 2 becomes $\mathcal{O}(V^3 N)$. Another example is to train Eq. 2 with stochastic optimization methods, where the complexity of Eq. 2 on the full dataset can be reduced to $\mathcal{O}(B^2 M N)$, where $B$ is the mini-batch size. Running stochastic optimization will then take $\mathcal{O}(B^2 M N K)$, where $K$ is the number of optimization epochs.

## C  OBJECTIVE FUNCTIONS

In §3, we presented NLL and KL divergence as objectives. Below we derive the KL divergence between a regular multivariate Gaussian and a degenerate multivariate Gaussian, which is the case for most of our matching data settings in §4.1: the number of matching data points is greater than the number of training tasks. In the end of this section, we introduce a new objective function, combining NLL and KL, interpreted as MAP with a data-dependent prior.

**KL divergence for a degenerate multivariate Gaussian**  Eq. 4 of §3.2 gives the KL divergence between two Gaussians in the non-degenerate case. In practice, when we minimize Eq. 4, we can simply remove the constants and do the following

$$
\begin{aligned}
\hat{\mu}, \hat{k}, \hat{\sigma}^2 &= \arg\min_{\mu,k,\sigma^2} \mathcal{D}_{\mathrm{KL}}\left(\mathcal{N}(\tilde{\boldsymbol{\mu}}, \tilde{K}), \mathcal{N}(\boldsymbol{\mu}, K)\right) \\
&= \arg\min_{\mu,k,\sigma^2} \frac{1}{2}\left(\mathrm{tr}(K^{-1}\tilde{K}) + (\boldsymbol{\mu} - \tilde{\boldsymbol{\mu}})^\top K^{-1}(\boldsymbol{\mu} - \tilde{\boldsymbol{\mu}}) + \ln\frac{|K|}{|\tilde{K}|} - M\right) \\
&= \arg\min_{\mu,k,\sigma^2} \mathrm{tr}(K^{-1}\tilde{K}) + (\boldsymbol{\mu} - \tilde{\boldsymbol{\mu}})^\top K^{-1}(\boldsymbol{\mu} - \tilde{\boldsymbol{\mu}}) + \ln|K|.
\end{aligned} \tag{6}
$$

Here the variables we care about, $\mu, k, \sigma^2$, only appear in mean vector $\boldsymbol{\mu}$ and covariance matrix $K$ over the matching data. Even if the sample mean and covariance estimate $\mathcal{N}(\tilde{\boldsymbol{\mu}}, \tilde{K})$ is degenerate, the optimization objective stays the same as reflected by the derivation below.

If $\mathcal{N}(\tilde{\boldsymbol{\mu}}, \tilde{K})$ is degenerate, its base measure is at most $N$-dimensional rather than $M$-dimensional, given that there exists a full rank matrix $A \in \mathbb{R}^{M \times R}$ such that $\tilde{K} = AA^\top$ ($R \le N$). Note that $M$ is the number of matching data points, $N$ the number of training tasks and $R$ is the rank of matrix $A$ and $\tilde{K}$. The KL divergence $D_{\mathrm{KL}}\left(\mathcal{N}(\tilde{\boldsymbol{\mu}}, \tilde{K}), \mathcal{N}(\boldsymbol{\mu}, K)\right)$ is not well-defined because the base measure of $\mathcal{N}(\tilde{\boldsymbol{\mu}}, \tilde{K})$ is different from the base measure of $\mathcal{N}(\boldsymbol{\mu}, K)$, given $K$ is full-rank. However, it is still possible to derive a pseudo KL divergence $D_{\mathrm{KL}}^*\left(\mathcal{N}(\tilde{\boldsymbol{\mu}}, \tilde{K}), \mathcal{N}(\boldsymbol{\mu}, K)\right)$ as below.

Let the degenerate Gaussian be $p(x) = \mathcal{N}(\tilde{\boldsymbol{\mu}}, \tilde{K}) = |2\pi\tilde{K}|_*^{-\frac{1}{2}}\exp\left(-\frac{1}{2}(x - \tilde{\boldsymbol{\mu}})\tilde{K}^+(x - \tilde{\boldsymbol{\mu}})^\top\right)$ and the non-degenerate one be $q(x) = \mathcal{N}(\boldsymbol{\mu}, K)$, where $|\cdot|_*$ is the pseudo-determinant and $\tilde{K}^+$ the pseudo-inverse of $\tilde{K}$. We define the support of distribution $p$ as $S(p) = \{\tilde{\boldsymbol{\mu}} + \tilde{K}^{\frac{1}{2}}v \mid v \in \mathbb{R}^M\}$. The pseudo KL divergence between $p(x)$ and $q(x)$ now becomes

$$
\begin{aligned}
&\mathcal{D}_{\mathrm{KL}}^*\left(\mathcal{N}(\tilde{\boldsymbol{\mu}}, \tilde{K}), \mathcal{N}(\boldsymbol{\mu}, K)\right) \\
&= \int_{S(p)} p(x)\left(\ln p(x) - \ln q(x)\right) \\
&= -\frac{1}{2}\int_{S(p)} p(x)\left(\ln|2\pi\tilde{K}|_* - \ln|2\pi K| + (x - \tilde{\boldsymbol{\mu}})^\top\tilde{K}^+(x - \tilde{\boldsymbol{\mu}}) - (x - \boldsymbol{\mu})^\top K^{-1}(x - \boldsymbol{\mu})\right) \\
&= \frac{1}{2}\left((M - R)\ln 2\pi + \ln\frac{|K|}{|A^\top A|} - \mathbb{E}_p[\mathrm{tr}(\tilde{K}^+(x - \tilde{\boldsymbol{\mu}})(x - \tilde{\boldsymbol{\mu}})^\top) + \mathrm{tr}(K^{-1}(x - \boldsymbol{\mu})(x - \boldsymbol{\mu})^\top)]\right) \\
&= \frac{1}{2}\left((M - R)\ln 2\pi + \ln\frac{|K|}{|A^\top A|} - \mathrm{tr}(\tilde{K}^+\tilde{K}) + \mathbb{E}_p[\mathrm{tr}(K^{-1}(x - \boldsymbol{\mu})(x - \boldsymbol{\mu})^\top)]\right) \\
&= \frac{1}{2}\left((M - R)\ln 2\pi + \ln\frac{|K|}{|A^\top A|} - \mathrm{tr}(AA^+) + \mathbb{E}_p[\mathrm{tr}(K^{-1}(x - \tilde{\boldsymbol{\mu}})(x - \tilde{\boldsymbol{\mu}})^\top + K^{-1}(2x\tilde{\boldsymbol{\mu}}^\top - 2x\boldsymbol{\mu}^\top - \tilde{\boldsymbol{\mu}}\tilde{\boldsymbol{\mu}}^\top + \boldsymbol{\mu}\boldsymbol{\mu}^\top))]\right) \\
&= \frac{1}{2}\left((M - R)\ln 2\pi + \ln\frac{|K|}{|A^\top A|} - R + \mathrm{tr}(K^{-1}\tilde{K} + K^{-1}(\tilde{\boldsymbol{\mu}} - \boldsymbol{\mu})(\tilde{\boldsymbol{\mu}} - \boldsymbol{\mu})^\top)\right) \\
&= \frac{1}{2}\left(\mathrm{tr}(K^{-1}\tilde{K}) + (\boldsymbol{\mu} - \tilde{\boldsymbol{\mu}})^\top K^{-1}(\boldsymbol{\mu} - \tilde{\boldsymbol{\mu}}) + \ln|K| - \ln|A^\top A| - R + (M - R)\ln 2\pi\right).
\end{aligned} \tag{7}
$$

If the covariance matrix $\tilde{K}$ is in fact full rank, i.e. $R = M$, pseudo KL in Eq. 7 then recovers the KL divergence $\mathcal{D}_{\mathrm{KL}}\left(\mathcal{N}(\tilde{\boldsymbol{\mu}}, \tilde{K}), \mathcal{N}(\boldsymbol{\mu}, K)\right)$ in Eq. 4 for non-degenerate Gaussians. If we were to minimize this pseudo KL divergence $\mathcal{D}_{\mathrm{KL}}^*\left(\mathcal{N}(\tilde{\boldsymbol{\mu}}, \tilde{K}), \mathcal{N}(\boldsymbol{\mu}, K)\right)$, we still would get the minimization task in Eq. 6. However, pseudo KL divergence $\mathcal{D}_{\mathrm{KL}}^*$ does not satisfy properties including non-negativity. In practice, we may also choose to add small epsilon values to the diagonal terms of both $K$ and $\tilde{K}$ to make $\tilde{K}$ "less degenerate" and enable the existence of $D_{\mathrm{KL}}\left(\mathcal{N}(\tilde{\boldsymbol{\mu}}, \tilde{K}), \mathcal{N}(\boldsymbol{\mu}, K)\right)$.

**Combining marginal likelihood and distance metrics**    It is also possible to additively combine the KL divergence with the negative log marginal likelihood objective, and treat this distance as a regularizer. In the case of KL divergence, it is equivalent to adding a data-dependent prior on the GP itself: $\boldsymbol{\mu}, K \sim \exp(-\lambda \mathcal{D}_{\text{KL}}(\mathcal{N}(\tilde{\boldsymbol{\mu}}, \tilde{K}), \mathcal{N}(\boldsymbol{\mu}, K)))/Z$ for some normalization constant $Z$, and the posterior is

$$p(\mu, k, \sigma^2 \mid D_N; \tilde{\boldsymbol{\mu}}, \tilde{K}) \propto p(D_N \mid \mu, k, \sigma^2) \exp(-\lambda \mathcal{D}_{\text{KL}} \left( \mathcal{N}(\tilde{\boldsymbol{\mu}}, \tilde{K}), \mathcal{N}(\boldsymbol{\mu}, K) \right)). \qquad (8)$$

We can then obtain an MAP estimation for the unknown functions and variables $\mu, k, \sigma^2$.

## D    DETAILS OF REGRET BOUNDS

Theorem 1 is a direct result of Theorem 16 in Wang et al. (2018b). The only subtle difference is that we adopted a biased estimate of the covariance matrix, a factor of $\frac{N}{N-1}$ different from the unbiased estimate. But this can be corrected in the acquisition functions. We provide more details below.

**Proposition 2.** *For any $M, d, N \in \mathbb{Z}^+, \boldsymbol{x} \in \mathbb{R}^{M \times d}, \boldsymbol{\mu} \in \mathbb{R}^M, V \in \mathbb{R}^{N \times M}$ and $K = V^\top V$, there exists a Gaussian process $\mathcal{GP}(\hat{\mu}, \hat{k})$ such that $\mathcal{D}_{KL} \left( \mathcal{N}(\boldsymbol{\mu}, K), \mathcal{N}(\hat{\mu}(\boldsymbol{x}), \hat{k}(\boldsymbol{x})) \right) \equiv 0$.*

Proposition 2 is easy to show. We can train a simple memory based model for mean function $\hat{\mu}$ and kernel $\hat{k}$. The model stores each element of vector $\boldsymbol{\mu}$ and matrix $K$ at the corresponding locations in input $\boldsymbol{x}$. When making a prediction on a new input $x' \in \mathbb{R}^d$, the model simply retrieves the values of the closest element in $\boldsymbol{x}$. Given Proposition 2, a regret bound follows (Wang et al., 2018b).

By Proposition 2, we are able to obtain a $\mathcal{GP}(\mu, k)$ where $\mu(\boldsymbol{x})$ and $k(\boldsymbol{x})$ are equal to the sample mean and covariance on a matching dataset $(\boldsymbol{x}, \boldsymbol{y})$, following the notations in §3.2. Hence, our problem setup is consistent with the case with discrete input space in Wang et al. (2018b). The following theorem is a rewrite of Theorem 16 in Wang et al. (2018b), taking into account our biased estimators.

**Theorem 3.** *Assume there exist constant $c \geq \max_{x \in \mathfrak{X}} k(x)$ and a training dataset is available whose size is $N \geq 4 \log \frac{6}{\delta} + T + 2$. Define*

$$\iota_{t-1} = \sqrt{\frac{6 \left( N - 3 + t + 2\sqrt{t \log \frac{6}{\delta}} + 2 \log \frac{6}{\delta} \right)}{\delta N (N - t - 1)}}, \; b_{t-1} = \frac{1}{N-t} \log \frac{6}{\delta}, \; \text{for any } t \in [T],$$

*and $\rho_T = \max\limits_{A \in \mathfrak{X}, |A| = T} \frac{1}{2} \log |\boldsymbol{I} + \sigma^{-2} k(A)|$. Then, with probability at least $1 - \delta$, the best-sample simple regret in $T$ iterations of meta BO with GP-UCB that uses*

$$\zeta_t = \frac{\left( 6N(N - 3 + t + 2\sqrt{t \log \frac{6}{\delta}} + 2 \log \frac{6}{\delta})/(\delta N(N - t - 1)) \right)^{\frac{1}{2}} + (2N \log(\frac{3}{\delta}))^{\frac{1}{2}}}{\left( (N-1)(1 - 2(\frac{1}{N-t} \log \frac{6}{\delta})^{\frac{1}{2}}) \right)^{\frac{1}{2}}} \qquad (9)$$

*as its hyperparameter in $\alpha_{t-1}^{GP\text{-}UCB}(x) = \hat{\mu}_{t-1}(x) + \zeta_t \hat{k}_{t-1}(x)^{\frac{1}{2}}$ satisfies*

$$r_T^{GP\text{-}UCB} \leq \eta^{GP\text{-}UCB} \sqrt{\frac{2c\rho_T}{T \log(1 + c\sigma^{-2})} + \sigma^2} - \frac{(2 \log(\frac{3}{\delta}))^{\frac{1}{2}} \sigma^2}{\sqrt{c + \sigma^2}},$$

*where $\eta^{GP\text{-}UCB} = (\frac{\iota_{T-1} + (2 \log(\frac{3}{\delta}))^{\frac{1}{2}}}{\sqrt{1 - 2\sqrt{b_{T-1}}}} \sqrt{1 + 2\sqrt{b_{T-1}} + 2b_{T-1}} + \iota_{T-1} + (2 \log(\frac{3}{\delta}))^{\frac{1}{2}})$.*

*With probability at least $1 - \delta$, the best-sample simple regret in T iterations of meta BO with PI that uses $\hat{f}^* \geq \max_{x \in \mathfrak{X}} f(x)$ as its target value satisfies*

$$r_T^{PI} < \eta^{PI} \sqrt{\frac{2c\rho_T}{T \log(1 + c\sigma^{-2})} + \sigma^2} - \frac{(2 \log(\frac{3}{2\delta}))^{\frac{1}{2}} \sigma^2}{2\sqrt{c + \sigma^2}},$$

*where $\eta^{PI} = (\frac{\hat{f}^* - \mu_{\tau-1}(x_*)}{\sqrt{k_{\tau-1}(x_*) + \sigma^2}} + \iota_{\tau-1}) \sqrt{\frac{1 + 2b_{\tau-1}^{\frac{1}{2}} + 2b_{\tau-1}}{1 - 2b_{\tau-1}^{\frac{1}{2}}}} + \iota_{\tau-1} + (2 \log(\frac{3}{2\delta}))^{\frac{1}{2}}, \; \tau = \arg\min_{t \in [T]} k_{t-1}(x_t)$.*

The proof can be found in Wang et al. (2018b). Theorem 1 is a condensed version of Theorem 3.

While Theorem 3 provides us with some understanding of HyperBO in a specific setting, in practice, we need to query in a continuous input space that goes beyond the finite set of points present in the training dataset. It may or may not be possible to obtain data on a wide range of tasks to ensure $N \geq 4 \log \frac{6}{\delta} + T + 2$. In fact, in all of our experiment, this criterion on number of tasks is not satisfied. However, we still obtained good performance.

# E  EXPERIMENT DETAILS AND MORE RESULTS

In this section, we provide more detailed setups and empirical results on the impact of objective functions and acquisition functions in HyperBO. All experiment setups are the same as §E.2.1: offline and holding out related tasks.

## E.1  BASELINES

Our first set of baselines include those that *do not* use information from training tasks:

- Rand: Random search in the corresponding scaled space in Tab. 2.
- STBO: Single task BO with a constant mean function, Matern32 kernel and PI acquisition function (same as above). Every BO iteration, STBO optimizes the GP hyperparameters via marginal likelihood on data of the test task. This implementation corresponds to the basic off-the-shelf BO setups.
- STBOH: Single task GP-UCB (coefficient=1.8) with constant mean, Matern52 kernel and *hand-tuned* prior on hyper-parameters including UCB coefficient. Specifically, log amplitude follows Normal(-1, 1), log length scale (one per input parameter) follows Normal(0,1), and log observation noise variance follows Normal(-6, 3). The hyperparameters are post-processed by tensorflow-probability's `SoftClip` bijector to constrain the values between 1-st and 99-th quantiles. These prior distributions were manually tuned to obtain reasonable convergence rates on 24 analytic functions in COCO (Hansen et al., 2021). The GP parameters are then optimized via maximum marginal likelihood every BO iteration.

For multi-task BO baselines, we included scalable methods that replace the GP with a regression model that can be trained using SGD and thus scales linearly in the number of observations. Following the multi-task setup of Springenberg et al. (2016), we jointly trained a 5-dimensional embedding of each task, which was then added to the input of the following two models.

- MIMO: We trained an ensemble of feedforward neural networks with shared subnetworks (Havasi et al., 2020). We used 1 shared dense layer of size 10 and 2 unshared layers of size 10. We used tanh activation based on (Snoek et al., 2015, Figure 2). The network has one output unit with linear activation and another with $\text{softmax}(10^{-4}, 1)$ activation, corresponding respectively to the mean and standard deviation parameters of a normal distribution. We trained for 1000 epochs using the Adam optimizer with learning rate $10^{-4}$ and batch size 64.
- RFGP: We used the open-source implementation of approximate GP by Liu et al. (2020). We trained for 1000 epochs using the Adam optimizer with learning rate $10^{-3}$ and batch size 64.

All methods share the same input and output warping. The input warping is done according to the scaling function in Tab. 2: $\eta \leftarrow \log \eta, 1 - \beta \leftarrow \log(1 - \beta)$. The output warping is done for the best validation error rate $r \leftarrow -\log(r + 10^{-10})$.

## E.2  OFFLINE BO EXPERIMENTS

In all offline experiments, we ran offline BO on the data from the test task starting from zero initial data from this task. Each method was repeated 5 times with different random seeds to initialize its model. We ran all methods without de-duplication to best simulate online BO. We evaluate on regret on error rate which denotes the simple regret on the finite set of data points in each tuning sub-dataset.

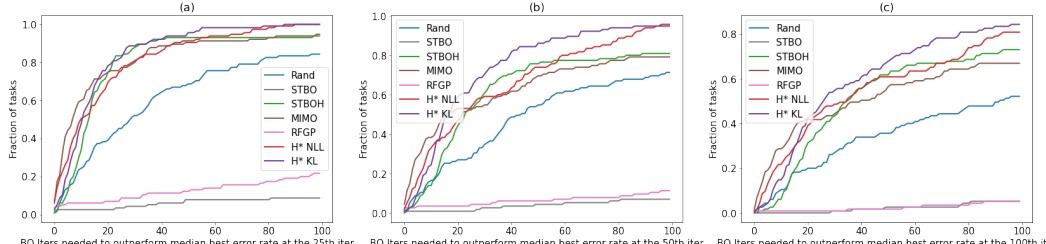

Figure 6: Performance profiles for outperforming the median of best error rates at the (a) 25th BO iteration, (b) 50th BO iteration and (c) 100th BO iteration.

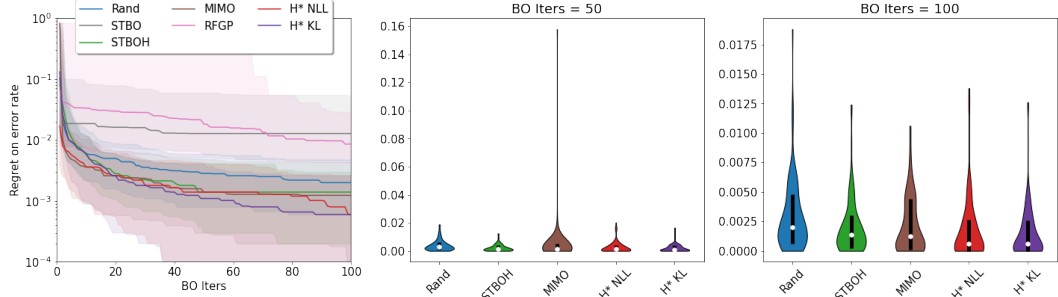

Figure 7: The left most is a summary of the BO convergence of all methods: the median and 20/80 percentiles of the regrets on error rates over 115 BO runs: 23 tasks and each with 5 repeats of different random seeds. We also show violin plots on its two vertical slices at 50th and 100th iteration, where the white dot is the median and the black line is the 20/80 percentile. Overall, HyperBO methods H* NLL and H* KL are able to achieve the lowest regret on error rate on the majority of tasks.

### E.2.1   HOLDING OUT RELEVANT TASKS

We first conducted experiments in a setting where a new task dataset is presented, and a BO method is trying to tune the optimizer hyperparameters for a selected model on that task dataset. A training dataset for meta BO is composed of at most 18 tuning sub-datasets on training tasks that do not involve the same task dataset as the test task. All methods then proceed to solve the test task on the new task dataset. Fig. 6 shows *performance profiles* of the BO methods described in §4.2. The performance profiles show the fraction of all test tasks that each method is able to outperform a baseline criterion at each BO iteration.[6] We chose the criterion to be the median of best error rates achieved by all methods at 3 different BO iterations: 25th, 50th or 100th. The larger the fraction of tasks at each BO iteration, the better the method is. From all 3 criteria, we can see that MIMO is able to outperform other methods in the beginning 10 to 20 BO iterations, but its leading position soon gets surpassed by HyperBO (H* NLL and H* KL) and STBOH. HyperBO methods are gaining a similar if not larger fraction than the best alternative, STBOH, throughout BO iterations. Fig. 6 (c) has the most stringent performance criterion, and it shows that HyperBO with the KL objective outperforms HyperBO with the NLL objective in this set of experiments with a small margin. And both methods in HyperBO are doing considerably better than others.

Fig. 7 illustrates the BO convergence curves of all competing methods, together with the vertical slices at the 50th and 100th iterations. RFGP and STBO are both falling much behind Random search. STBO trains the GP on the data that the GP suggests to query, which creates a loop that could be harmful for data acquisition. Optimizing the marginal data likelihood on at most 100 datapoints in fact may not lead to a better model than random initialization (see Tab. 5 in §F). Surprisingly, RFGP, though equipped with the tuning dataset and initially reached some good values, performed similarly to STBO in the end. Clearly, the contextual information learned by RFGP did not generalize to a new task. On the other hand, MIMO is able to obtain a slightly better error rate than STBOH.

Fig. 6 and Fig. 7 both show that learning the GP prior through data like what HyperBO does is performing much better than other meta BO methods, and it is a more principled and effective

---

[6]We show performance relative to a baseline because of varying scales across tasks.

approach to obtain the GP prior than hand-tuning. As a reference, we include Tab. 3 which shows the task-wise best validation error rates obtained by the top 5 methods in 100 BO iterations.

Table 3: The mean and standard error of best validation error rates (%) for each test task in the offline optimizer hyperparameter tuning experiments. Meta BO methods including MIMO and HyperBO variants (H* NLL and H* KL) have access to training tasks that do not share the same task dataset as the test task. We show results of the top 5 methods, and we highlight the lowest error rates in bold.

| | Rand | STBOH | MIMO | H* NLL | H* KL |
|---|---|---|---|---|---|
| WMT XFormer 64 | $34.27 \pm 0.16$ | $34.15 \pm 0.15$ | $34.29 \pm 0.16$ | $\mathbf{33.94 \pm 0.01}$ | $33.99 \pm 0.03$ |
| Uniref50 Transformer 128 | $79.06 \pm 0.04$ | $78.92 \pm 0.12$ | $78.93 \pm 0.11$ | $\mathbf{78.64 \pm 0.00}$ | $78.74 \pm 0.09$ |
| LM1B Transformer 2048 | $61.96 \pm 0.03$ | $61.95 \pm 0.04$ | $61.95 \pm 0.01$ | $\mathbf{61.83 \pm 0.01}$ | $\mathbf{61.82 \pm 0.01}$ |
| SVHN WRN 1024 | $3.99 \pm 0.04$ | $4.05 \pm 0.10$ | $\mathbf{3.82 \pm 0.04}$ | $4.11 \pm 0.04$ | $4.06 \pm 0.02$ |
| SVHN WRN 256 | $3.71 \pm 0.01$ | $3.72 \pm 0.02$ | $\mathbf{3.62 \pm 0.02}$ | $3.79 \pm 0.01$ | $3.77 \pm 0.02$ |
| ImageNet ResNet50 256 | $23.03 \pm 0.07$ | $22.66 \pm 0.07$ | $22.69 \pm 0.06$ | $22.57 \pm 0.02$ | $\mathbf{22.55 \pm 0.02}$ |
| ImageNet ResNet50 512 | $23.02 \pm 0.11$ | $22.74 \pm 0.05$ | $22.99 \pm 0.05$ | $\mathbf{22.65 \pm 0.02}$ | $22.75 \pm 0.03$ |
| MNIST CNNPoolTanh 2048 | $0.55 \pm 0.01$ | $\mathbf{0.53 \pm 0.01}$ | $0.52 \pm 0.01$ | $0.53 \pm 0.01$ | $0.54 \pm 0.01$ |
| MNIST CNNPoolTanh 256 | $0.51 \pm 0.01$ | $0.48 \pm 0.01$ | $\mathbf{0.46 \pm 0.00}$ | $0.46 \pm 0.01$ | $0.46 \pm 0.01$ |
| MNIST CNNPoolReLU 2048 | $0.69 \pm 0.01$ | $0.73 \pm 0.02$ | $0.66 \pm 0.01$ | $\mathbf{0.64 \pm 0.01}$ | $0.65 \pm 0.01$ |
| MNIST CNNPoolReLU 256 | $0.51 \pm 0.01$ | $0.55 \pm 0.03$ | $0.52 \pm 0.01$ | $\mathbf{0.48 \pm 0.00}$ | $0.49 \pm 0.00$ |
| MNIST CNNReLU 2048 | $1.14 \pm 0.03$ | $1.20 \pm 0.09$ | $1.11 \pm 0.02$ | $\mathbf{1.06 \pm 0.00}$ | $1.08 \pm 0.01$ |
| MNIST CNNReLU 256 | $1.09 \pm 0.02$ | $1.06 \pm 0.01$ | $1.08 \pm 0.02$ | $\mathbf{1.03 \pm 0.00}$ | $\mathbf{1.03 \pm 0.00}$ |
| Fashion CNNPoolTanh 2048 | $7.14 \pm 0.06$ | $7.10 \pm 0.05$ | $\mathbf{7.05 \pm 0.06}$ | $7.03 \pm 0.03$ | $7.16 \pm 0.02$ |
| Fashion CNNPoolTanh 256 | $6.51 \pm 0.03$ | $6.67 \pm 0.18$ | $6.41 \pm 0.07$ | $6.38 \pm 0.02$ | $\mathbf{6.28 \pm 0.01}$ |
| Fashion CNNPoolReLU 2048 | $7.47 \pm 0.02$ | $7.48 \pm 0.04$ | $7.52 \pm 0.06$ | $\mathbf{7.42 \pm 0.03}$ | $7.53 \pm 0.04$ |
| Fashion CNNPoolReLU 256 | $6.78 \pm 0.04$ | $6.74 \pm 0.01$ | $6.89 \pm 0.06$ | $6.79 \pm 0.04$ | $\mathbf{6.70 \pm 0.01}$ |
| Fashion CNNReLU 2048 | $7.70 \pm 0.03$ | $\mathbf{7.47 \pm 0.09}$ | $7.64 \pm 0.06$ | $7.57 \pm 0.01$ | $7.56 \pm 0.02$ |
| Fashion CNNReLU 256 | $7.70 \pm 0.04$ | $7.46 \pm 0.11$ | $7.83 \pm 0.05$ | $7.44 \pm 0.12$ | $\mathbf{7.25 \pm 0.05}$ |
| CIFAR100 WRN 2048 | $21.28 \pm 0.27$ | $\mathbf{20.78 \pm 0.19}$ | $20.94 \pm 0.13$ | $21.26 \pm 0.23$ | $20.98 \pm 0.24$ |
| CIFAR100 WRN 256 | $19.17 \pm 0.19$ | $19.02 \pm 0.03$ | $19.15 \pm 0.06$ | $19.07 \pm 0.04$ | $\mathbf{18.98 \pm 0.01}$ |
| CIFAR10 WRN 2048 | $3.73 \pm 0.05$ | $\mathbf{3.43 \pm 0.07}$ | $3.40 \pm 0.04$ | $3.66 \pm 0.11$ | $\mathbf{3.43 \pm 0.07}$ |
| CIFAR10 WRN 256 | $2.84 \pm 0.04$ | $2.88 \pm 0.06$ | $2.84 \pm 0.05$ | $2.83 \pm 0.03$ | $\mathbf{2.80 \pm 0.02}$ |

To more precisely quantify HyperBO's advantage, we also computed how much faster HyperBO can get a better error rate than best alternatives, which can be different from task to task. We found that on average, on over 50% tasks, H* NLL is at least 2.86 times faster than best non-HyperBO alternatives; while on over 57% tasks, H* KL is at least 3.26 times faster than best non-HyperBO alternatives. Moreover, on over 73% tasks, H* NLL is at least 7.74 times faster than random search; and on over 75% tasks, H* KL is at least 6.07 times faster than random search.

### E.2.2 EFFECT OF NUMBER OF TRAINING TASKS

We now investigate the impact of number of training tasks on the performance of meta BO methods. In Fig 8 we show the BO simple regrets on tasks from Table 1 (except ImageNet ResNet50 2048) that use meta BO models trained on different number of training tasks. To analyze the performance of all methods on less-related tasks, we first remove training tasks that have the same task dataset as our current tuning task for testing, and then remove randomly selected training datasets from the rest.

HyperBO variants were able to reduce the simple regret as more training tasks are given. Interestingly, both H* NLL and H* KL are already slightly better than Rand and STBOH when they started off with only 3 training tasks. There are reasonable fluctuations in the results but overall the trend of regret is going down as the number of training tasks increases. MIMO also reduced regret when the number of tasks increased from 8 to 18. RFGP, however, fails to learn from training tasks possibly because it did not learn good task embeddings for GP regression models.

### E.2.3 EFFECT OF NUMBER OF DATA POINTS IN TRAINING TASKS

One remaining question is, how does $M_i$ in §2, the number of data points in each training tasks, affect the performance of meta BO methods. We analyze the impact of $M_i$ by removing a portion of all data that we have access to for each task. In particular, we set the percentage of remaining data to be $0.2\%, 0.5\%, 1\%, 3\%, 5\%, 10\%, 30\%, 50\%, 70\%, 90\%$. Remaining datapoints are selected uniformly randomly, which breaks the structure of matching data. Hence we do not include H* KL in this comparison, as H* KL only makes use of matching data.

Fig. 9 shows how the simple regret changes as the fraction of training data grows. Below 10% training data, we observe clear trend that more data lead to lower regret for both H* NLL and MIMO, and relatively no change for RFGP. We also found that the performance of HyperBO (H* NLL) does not change much as the fraction of training data increases from 5% to 90%. However, MIMO and RFGP

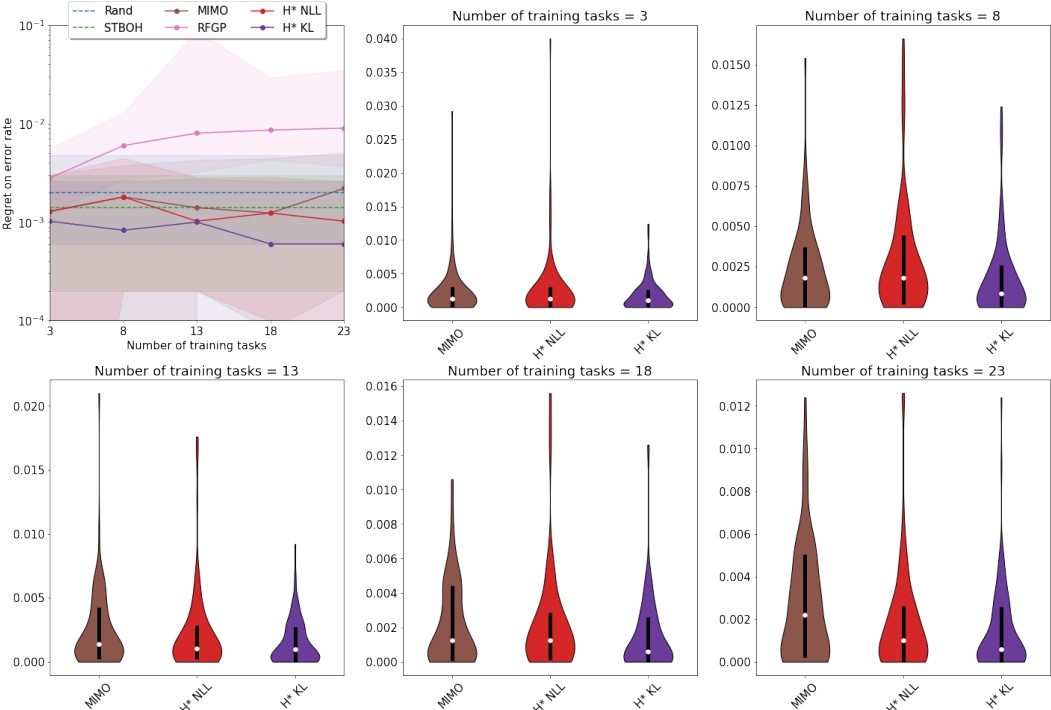

Figure 8: Aggregated BO results on 23 tasks (all in Table 1 except ImageNet ResNet50 2048 because of insufficient data) that uses models trained on 3 to 23 training tasks. Note that the models are never trained on the data from the test task that we run BO on. If the number of training tasks is less than 23, we first remove the tasks that involve the same task dataset as the test task and then remove others randomly until we reach the designated number of training tasks. The top left shows the median and 20/80 percentiles of regret on best validation error rate for each method. The rest are violin plots showing the regret for MIMO, H* NLL and H* KL, where white dots indicate the median and black lines the 20/80 percentiles.

suffers significantly from more data as the fraction of training data increases from 5% to 50%. It is not entirely clear why MIMO and RFGP have such behaviors. One conjecture is that neural network based Bayesian linear regression models may get too confident once the amount of data reaches a certain threshold. This means much less exploration if those models are used for BO.

### E.2.4 TRAINING ON ALL BUT ONE TASK

We also studied the case where meta BO approaches have access to both training tasks that do not use the same task dataset and training tasks that use the same task dataset but different model configurations. This is especially common when we do architecture search: we aim to find the best model and we are tuning the optimizer hyperparameters for a new machine learning model given tuning data of the same task dataset on some other models.

For this section only, we added a new baseline, MAF: We refer to the meta BO method from Volpp et al. (2020) as MAF (Meta Acquisition Function) to avoid confusion. MAF used reinforcement learning to learn an acquisition function modeled by a neural network over a set of transfer learning tasks. All MAF results were generated using the code from Volpp et al. (2020). See App. E.6 for experimental details. As MAF takes significantly longer to run than HyperBO and other methods, we only include its results for this section.

We carried out a series of leave-one-out experiments, where we picked one task as the BO test function and let meta BO methods train on the remaining tasks. In Fig. 10, we aggregated results from all 23 tasks to show the trend of how each method performs.

The conclusions are similar to those from §E.2.1. As expected, STBO without any tricks to avoid pitfalls of vanilla BO did not show very good results. We inspected its learned GP which mimicked a

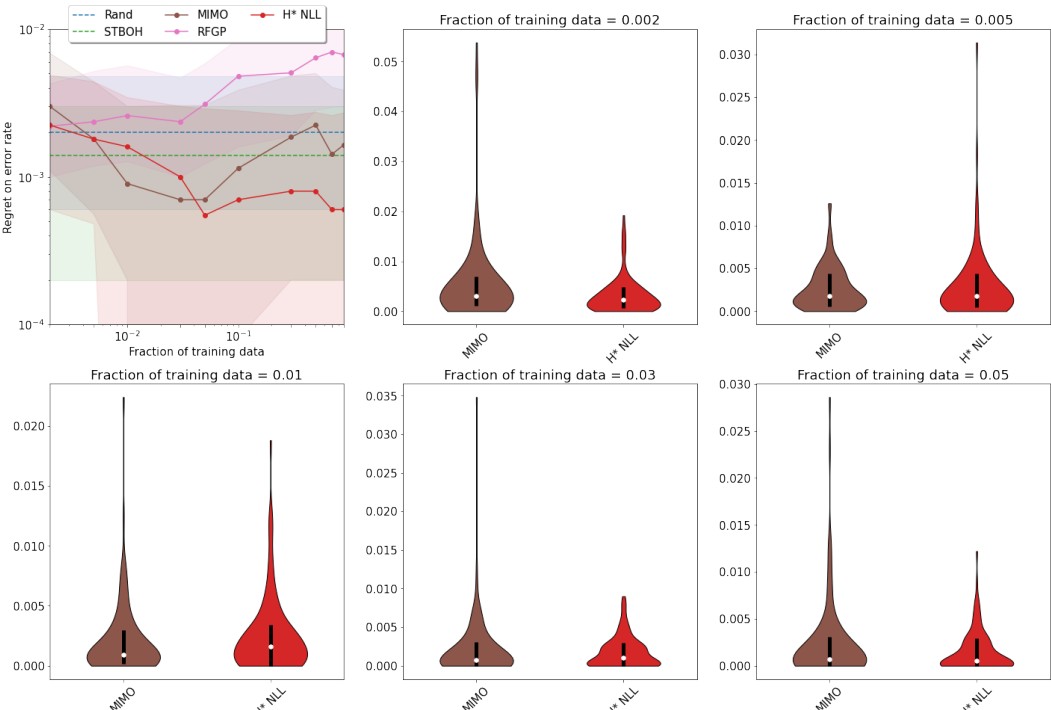

Figure 9: Aggregated BO results on 23 tasks (all in Table 1 except ImageNet ResNet50 2048 because of insufficient data) that uses models trained on $0.2\%$ to $90\%$ of data in each task. Note that the models are never trained on the data from the test task that we run BO on. The top left is the median and 20/80 percentiles of simple regret in log scale. The rest of the figures are simple regret violin plots for MIMO and H* NLL

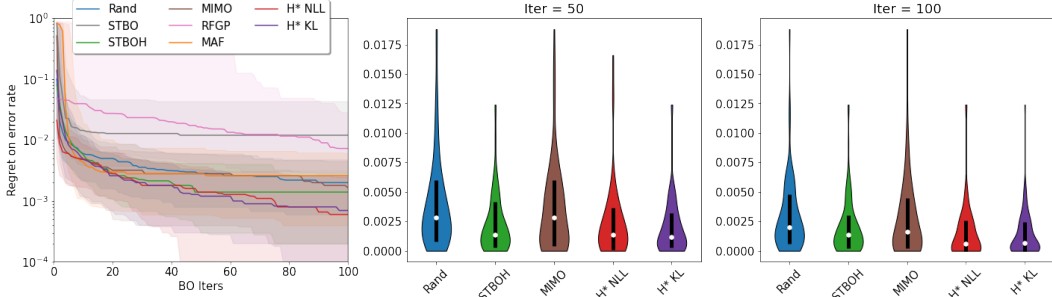

Figure 10: Aggregated leave-one-out BO convergence results on 23 tasks, each with 5 repeats using different random seeds. The left most is the median and 20/80 percentiles of the regrets on error rates. We also show violin plots on its two vertical slices at 50th and 100th iteration, where the white dot is the median and the black line is the 20/80 percentile.

Dirac function that is flat almost everywhere except some locations, and hence it got very confident that it landed at a good spot and lost its ability to explore.

STBOH, on the other hand, achieved very competitive results. This is because it used hand-tuned priors on all of its GP parameters, although they were tuned on somewhat different problems than the ones we consider. As part of the goals of meta learning, we would like to show that it is possible for meta BO methods to exceed or at least match STBOH.

Both HyperBO variants obtained better results than the hand-tuned STBOH. Especially in the beginning few BO iterations, it is able to locate much better hyperparameters than all other methods.

Tab. 4 presents mean and standard error of the best validation error rates achieved in 100 BO iterations on the 23 tasks. HyperBO and its variants were able to achieve the best performance on 20 out of 23 tasks. In Fig. 11, we show the optimization curves of 4 individual tasks that are considered most

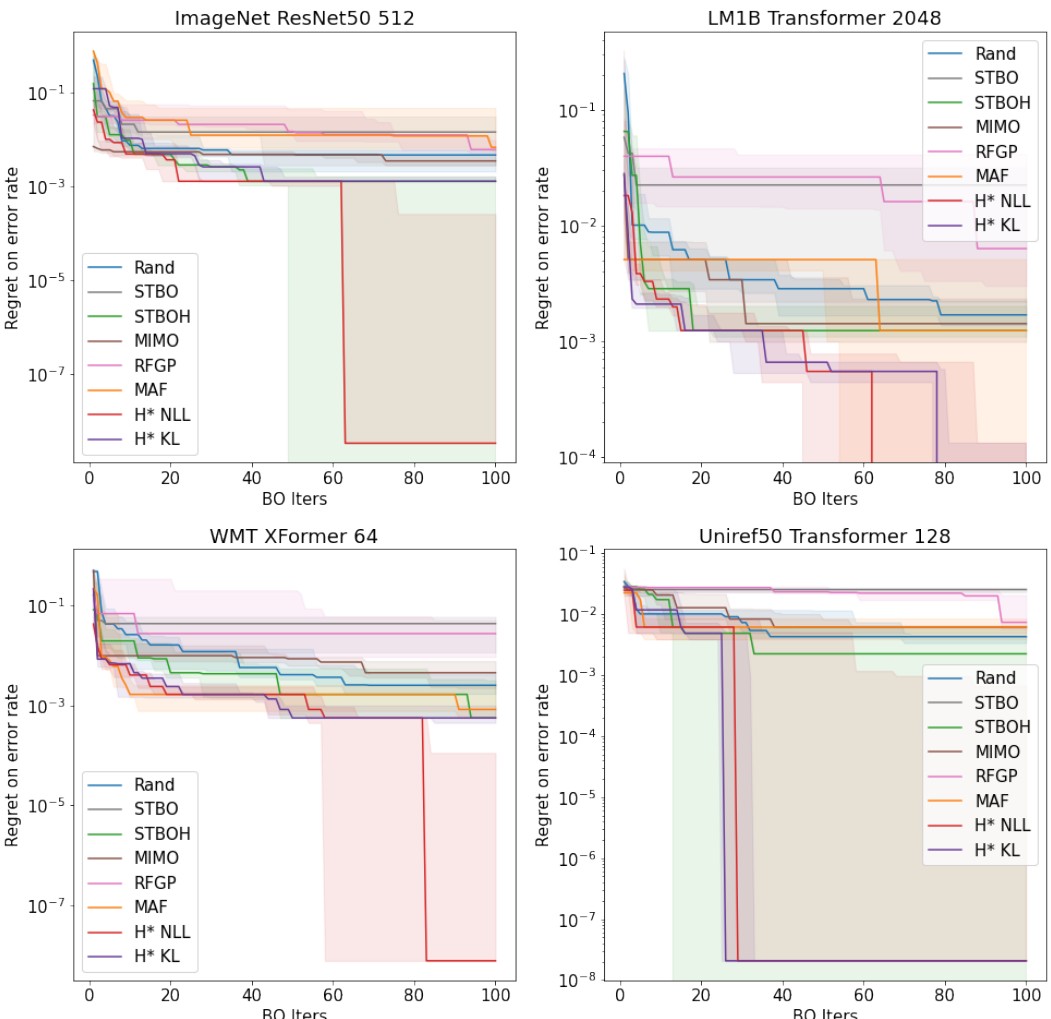

Figure 11: Leave-one-out log regret mean and standard deviation results on ImageNet ResNet50 512, LM1B Transformer 2048, WMT XFormer 64 and Uniref50 Transformer 128. All methods were repeated 5 times with different random seeds to initialize their models. In LM1B Transformer 2048, H* NLL and H* KL disappeared around 60 to 80 BO iterations because they reached 0 regret.

difficult because few similar task datasets are present in their training data. On all of these 4 difficult tasks, HyperBO identified good hyperparameters much sooner than its competitors.

### E.3 RESULTS ON ONLINE OPTIMIZER HYPERPARAMETER TUNING TASKS

Finally, we look into the online BO setting where we optimize over the full hypercube. In the online setting, some combinations of hyperparameters may be *infeasible* to evaluate. For example, an overly big learning rate may lead to divergence in gradients, in which case we do not obtain a valid model. To address this, we pre-process the function values to $[-2, 2)$ such that *infeasible* evaluations map to $-2$, while bad evaluations approach asymptotically to $-2$. More precisely, for each subdataset $D_{f_i}$, we applied for each successful $y \in \{y_{j^{(i)}}\}_{j=1}^{M_i}$ the following mapping:

$$y \leftarrow \frac{\text{softplus}(y - \overline{y})}{\text{softplus}(y_{\max} - \overline{y})} * 4 - 2$$

where $\overline{y}$ is the median of $\{y_{j^{(i)}}\}_{j=1}^{M_i}$.

Table 4: The mean and standard error of best validation error rates (%) for each test task in the offline leave-one-out experiments. We show results of the top 6 methods, and we highlight the lowest error rates in bold.

| | Rand | STBOH | MIMO | MAF | H* NLL | H* KL |
|---|---|---|---|---|---|---|
| WMT XFormer 64 | $34.27 \pm 0.16$ | $34.15 \pm 0.15$ | $34.40 \pm 0.13$ | $34.09 \pm 0.09$ | $\mathbf{33.91 \pm 0.01}$ | $33.97 \pm 0.02$ |
| Uniref50 Transformer 128 | $79.06 \pm 0.04$ | $78.92 \pm 0.12$ | $79.17 \pm 0.13$ | $79.34 \pm 0.27$ | $78.71 \pm 0.06$ | $\mathbf{78.64 \pm 0.00}$ |
| LM1B Transformer 2048 | $61.96 \pm 0.03$ | $61.95 \pm 0.04$ | $61.96 \pm 0.05$ | $62.02 \pm 0.10$ | $\mathbf{61.81 \pm 0.01}$ | $61.81 \pm 0.01$ |
| SVHN WRN 1024 | $3.99 \pm 0.04$ | $4.05 \pm 0.10$ | $\mathbf{3.83 \pm 0.04}$ | $4.10 \pm 0.09$ | $4.10 \pm 0.02$ | $4.08 \pm 0.01$ |
| SVHN WRN 256 | $3.71 \pm 0.01$ | $3.72 \pm 0.02$ | $\mathbf{3.65 \pm 0.01}$ | $3.69 \pm 0.03$ | $3.78 \pm 0.01$ | $3.72 \pm 0.03$ |
| ImageNet ResNet50 256 | $23.03 \pm 0.07$ | $22.66 \pm 0.07$ | $22.73 \pm 0.07$ | $26.44 \pm 1.98$ | $\mathbf{22.53 \pm 0.00}$ | $22.58 \pm 0.04$ |
| ImageNet ResNet50 512 | $23.02 \pm 0.11$ | $22.74 \pm 0.05$ | $23.01 \pm 0.05$ | $25.46 \pm 1.41$ | $\mathbf{22.65 \pm 0.02}$ | $22.79 \pm 0.03$ |
| MNIST CNNPoolTanh 2048 | $0.55 \pm 0.01$ | $\mathbf{0.53 \pm 0.01}$ | $\mathbf{0.53 \pm 0.01}$ | $\mathbf{0.52 \pm 0.01}$ | $0.59 \pm 0.02$ | $0.54 \pm 0.00$ |
| MNIST CNNPoolTanh 256 | $0.51 \pm 0.01$ | $0.48 \pm 0.01$ | $\mathbf{0.47 \pm 0.00}$ | $0.47 \pm 0.01$ | $0.46 \pm 0.01$ | $0.47 \pm 0.01$ |
| MNIST CNNPoolReLU 2048 | $0.69 \pm 0.01$ | $0.73 \pm 0.02$ | $0.67 \pm 0.02$ | $0.68 \pm 0.01$ | $\mathbf{0.64 \pm 0.00}$ | $0.70 \pm 0.03$ |
| MNIST CNNPoolReLU 256 | $0.51 \pm 0.01$ | $0.55 \pm 0.03$ | $0.50 \pm 0.01$ | $0.51 \pm 0.01$ | $\mathbf{0.49 \pm 0.00}$ | $\mathbf{0.49 \pm 0.00}$ |
| MNIST CNNReLU 2048 | $1.14 \pm 0.03$ | $1.20 \pm 0.09$ | $1.10 \pm 0.01$ | $1.17 \pm 0.02$ | $\mathbf{1.06 \pm 0.00}$ | $1.11 \pm 0.02$ |
| MNIST CNNReLU 256 | $1.09 \pm 0.02$ | $1.06 \pm 0.01$ | $1.08 \pm 0.02$ | $1.07 \pm 0.02$ | $\mathbf{1.03 \pm 0.00}$ | $1.04 \pm 0.01$ |
| Fashion CNNPoolTanh 2048 | $7.14 \pm 0.06$ | $7.10 \pm 0.05$ | $\mathbf{7.01 \pm 0.04}$ | $7.12 \pm 0.04$ | $7.00 \pm 0.04$ | $7.02 \pm 0.07$ |
| Fashion CNNPoolTanh 256 | $6.51 \pm 0.03$ | $6.67 \pm 0.18$ | $6.40 \pm 0.05$ | $6.47 \pm 0.03$ | $6.40 \pm 0.04$ | $\mathbf{6.34 \pm 0.04}$ |
| Fashion CNNPoolReLU 2048 | $\mathbf{7.47 \pm 0.02}$ | $7.48 \pm 0.04$ | $7.54 \pm 0.06$ | $7.63 \pm 0.04$ | $7.47 \pm 0.03$ | $7.47 \pm 0.02$ |
| Fashion CNNPoolReLU 256 | $6.78 \pm 0.04$ | $\mathbf{6.74 \pm 0.01}$ | $7.03 \pm 0.07$ | $6.84 \pm 0.05$ | $6.74 \pm 0.03$ | $6.81 \pm 0.05$ |
| Fashion CNNReLU 2048 | $7.70 \pm 0.03$ | $\mathbf{7.47 \pm 0.09}$ | $7.60 \pm 0.04$ | $40.40 \pm 17.80$ | $7.54 \pm 0.01$ | $7.57 \pm 0.02$ |
| Fashion CNNReLU 256 | $7.70 \pm 0.04$ | $7.46 \pm 0.11$ | $7.84 \pm 0.06$ | $24.13 \pm 14.54$ | $7.29 \pm 0.05$ | $\mathbf{7.25 \pm 0.05}$ |
| CIFAR100 WRN 2048 | $21.28 \pm 0.27$ | $\mathbf{20.78 \pm 0.19}$ | $21.75 \pm 0.15$ | $50.70 \pm 15.44$ | $21.22 \pm 0.23$ | $20.82 \pm 0.19$ |
| CIFAR100 WRN 256 | $19.17 \pm 0.19$ | $19.02 \pm 0.03$ | $19.12 \pm 0.04$ | $19.84 \pm 0.13$ | $\mathbf{19.00 \pm 0.00}$ | $19.04 \pm 0.05$ |
| CIFAR10 WRN 2048 | $3.73 \pm 0.05$ | $\mathbf{3.43 \pm 0.07}$ | $\mathbf{3.46 \pm 0.05}$ | $\mathbf{3.40 \pm 0.06}$ | $3.55 \pm 0.10$ | $\mathbf{3.43 \pm 0.05}$ |
| CIFAR10 WRN 256 | $2.84 \pm 0.04$ | $2.88 \pm 0.06$ | $2.89 \pm 0.06$ | $3.04 \pm 0.05$ | $2.82 \pm 0.03$ | $\mathbf{2.74 \pm 0.01}$ |

In this section, we set HyperBO variants and STBO to share exactly the same GP-UCB acquisition function as STBOH, MIMO and RFGP. The UCB coefficient for all methods is $1.8$. The variants of HyperBO are as follows:

- H* NLL: HyperBO with UCB as the acquisition function and negative log marginal likelihood (NLL) as the objective function.

- H* NLLKL: HyperBO with UCB as the acquisition function and NLL plus 10 times KL divergence on matching datapoints as the objective function. See §C for more details.

In Fig. 12, we include the online tuning results for selected tasks due to limited compute resources. We noticed that for some methods, e.g. STBO and MIMO, it is very difficult for them to recover from a "bad" datapoint. This is partly because predictions from these models are significantly tied to the initial observations. For example, STBO may overfit to the initial bad value and believe there are bad values in the entire search space. Nevertheless, in 7 out of 9 tasks, HyperBO methods performed the best among all methods being compared.

### E.4 IMPACT OF OBJECTIVE FUNCTIONS

Here we investigate how different objective functions in HyperBO can impact its performance. Besides NLL and KL, which are already described in details in §3, we also include NLL+KL, which corresponds to Eq. 8 with $\lambda = 10$. $\lambda = 10$ is an arbitrary choice, and one may find other better options to set $\lambda$ in Eq. 8.

Figure 13 shows the performance profiles and BO simple regret curves of NLL, KL and NLL+KL when HyperBO uses different acquisition functions. As comparisons, we also included the better performing baselines: Rand, STBOH and MIMO. While the ranks of performance by different objectives do vary depending on which acquisition function we used, it is clear that all HyperBO variants outperform the baselines. For EI and PI, the KL objective gives better performance, but NLL or NLL+KL might be preferred for UCB with coefficient 2, 3 or 4.

### E.5 IMPACT OF ACQUISITION FUNCTIONS

As explained briefly in §4, we used 5 acquisition functions in our experiments: the vannila EI method, PI with coefficient 0.1, $\alpha^{\text{PI}}\left(x; \mathcal{GP}(\hat{\mu}, \hat{k} \mid D_f)\right) = \frac{\hat{\mu}_{D_f}(x) - \max_t(y_t + 0.1)}{\hat{\sigma}_{D_f}(x)}$, and UCB with coefficient $\zeta = 2, 3$ and $4$ in $\alpha^{\text{UCB}}\left(x; \mathcal{GP}(\hat{\mu}, \hat{k} \mid D_f)\right) = \hat{\mu}_{D_f}(x) + \zeta\hat{\sigma}_{D_f}(x)$.

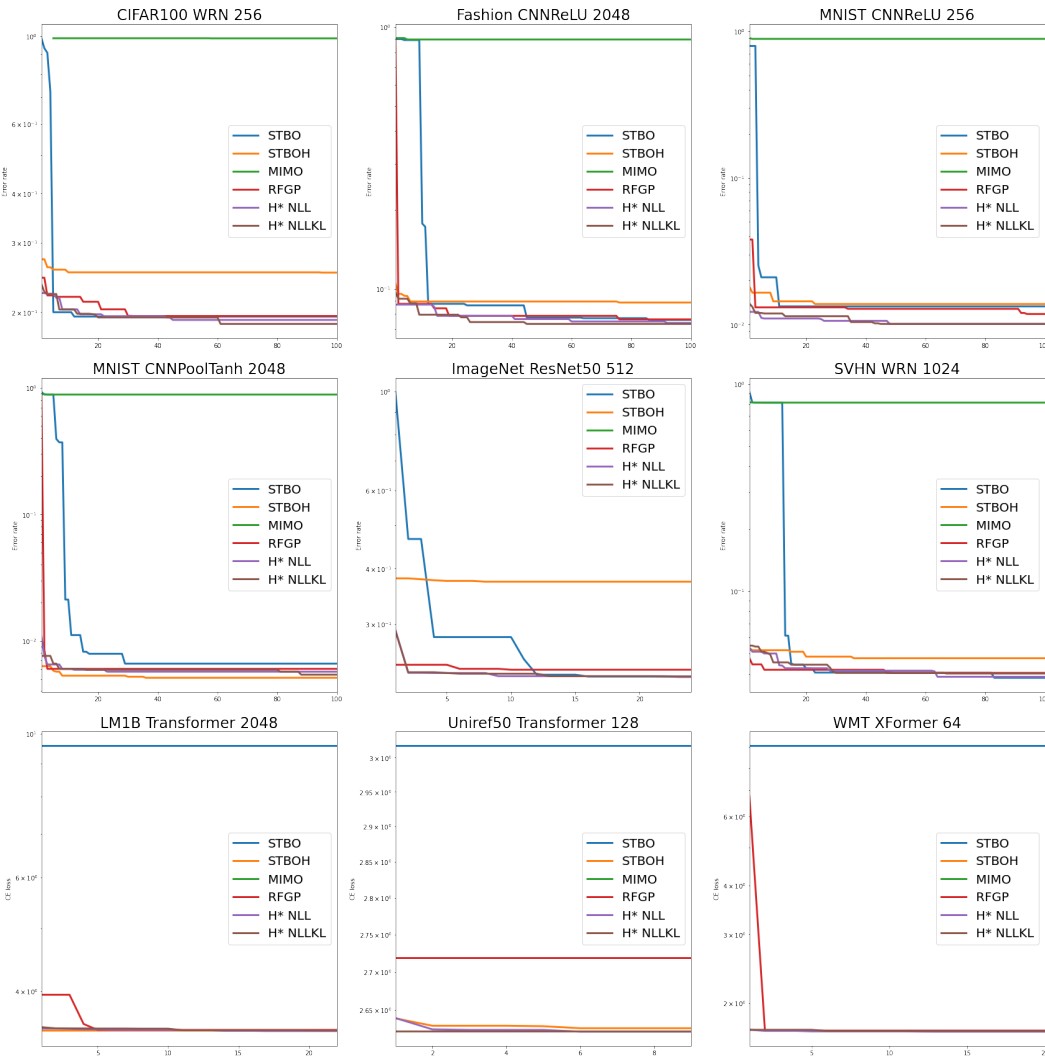

Figure 12: Results of running BO methods in the online setting on 9 different tasks. The image based tasks all use best validation error rate as objective while the text based tasks (LM1B, Uniref50 and WMT) use best validation ce loss. HyperBO methods achieved better results in 7 out of 9 tasks.

Our goal is to verify that HyperBO maintains good performance across different choices of acquisition functions. To do so, we investigated in the performance of HyperBO variants under different objectives. We avoid over cluttering the figures by only including STBOH as baseline, since it is roughly the best baseline according to the main results in Fig. 6.

As shown in Figure 14, HyperBO with EI and PI as acquisition functions perform relatively better than HyperBO with UCB variants. However, HyperBO with UCB3 can still be very competitive when it is coupled with NLL objective. Overall, HyperBO with all of the 5 acquisition function options outperforms the best performing baselines.

## E.6 MAF IMPLEMENTATION DETAILS

We compared to (Volpp et al., 2020) using the code and default hyperparameters provided by the authors.[7] This code assumes that each task is additionally accompanied by the optimal set of hyperparameters for the GP used to model the task (including the task used for evaluation). Following the MAF approach, we learned these hyperparameters using the GPY library, and provided them to

---

[7]https://github.com/boschresearch/MetaBO

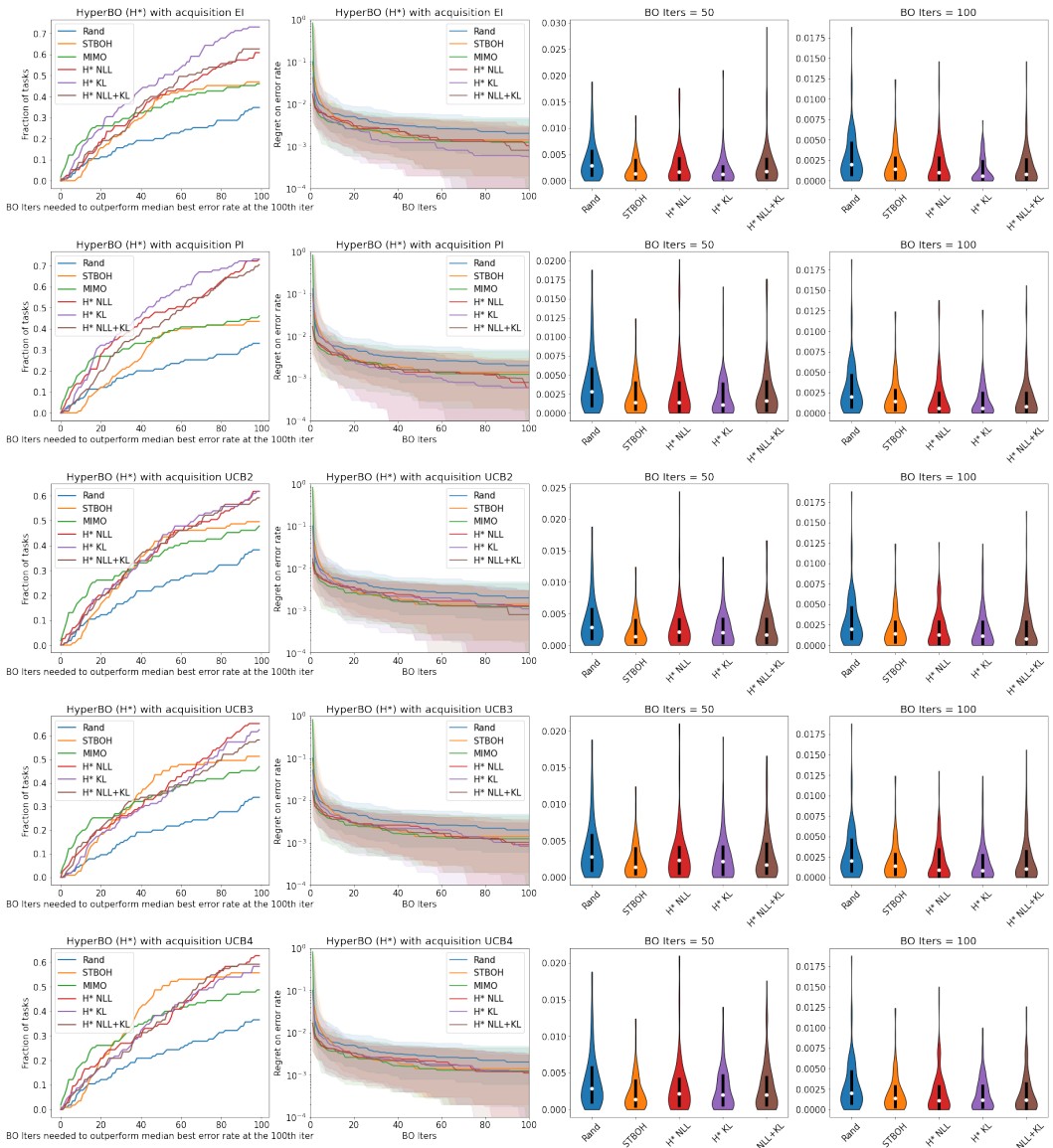

Figure 13: We compare the performance of 3 different objective functions in HyperBO under 5 settings of acquisition functions. For EI and PI, using KL as the objective for HyperBO is slightly better than NLL or NLL+KL. However, different conclusions can be drawn for UCB2, UCB3 and UCB4. Nevertheless, all HyperBO variants still outperform the best alternatives.

the MAF algorithm. Given that MAF takes significantly longer to run than HyperBO, each subdataset was evaluated using only one random seed.

Each neural acquisition function was trained for a total of 1000 iterations. As was done in (Volpp et al., 2020), we selected the optimal training iteration for the neural acquisition function by cross-validation on the transfer learning tasks; in this case, we randomly sampled 3 transfer learning task, and chose the training iteration with the lowest average simple regret.

Finally, to reuse the MAF code, we also had to ensure that (a) each subtask had the same number of evaluation points, and (b) that there were no duplicated hyperparameters. For this reason, we first removed all duplicate hyperparameters within each subdataset, then capped each subdataset to the first 1559 points (the size of the smallest sub-dataset).

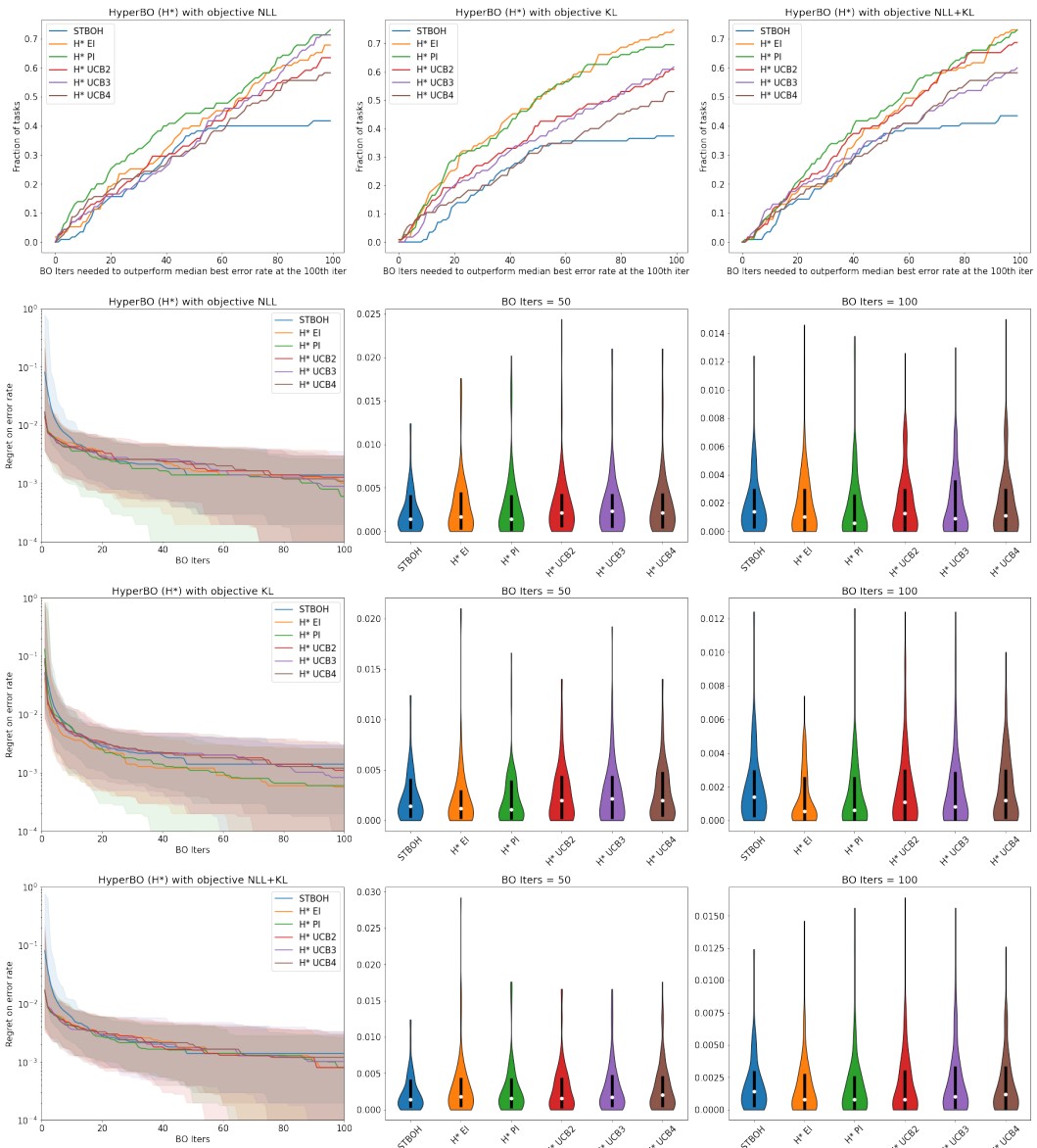

Figure 14: We compare the performance of 5 different acquisition functions under 3 settings of objectives in HyperBO. Overall, PI and EI outperform UCB with different coefficient values. But HyperBO with UCB variants still outperforms STBOH, which is roughly the best baseline according to the main results in Fig. 6.

# F  DISCUSSION

In this work, we focused on the question of how to make use of multi-task data to enable better Bayesian optimization. For the convenience of our investigation, we made simplifications such as sequential evaluations and a shared search space across tasks, although these are mostly unnecessary (see below). Our method also relies on an important assumption: functions of all tasks are *i.i.d.* samples from the same GP. In this section, we explore how reasonable the *i.i.d.* assumption is and discuss extensions to our work that would enable even more flexible uses.

**Assumption on *i.i.d.* GP samples.**  To get a better idea on how much our assumptions helped on training the GP, we compare NLLs associated with 23 tasks in §4.1 with models obtained via 3 scenarios:

   (a) No training: a randomly initialized model with no training;

   (b) Single task: models trained on 100 randomly selected data points of the test task;

   (c) H*: models trained on 18 irrelevant tasks selected in §E.2.2.

Here case (c) corresponds to the method HyperBO used for training a GP and case (b) corresponds to the model STBO can obtain with 100 initial observations. In Tab. 5, we show NLLs of these 3 methods on all tasks[8] and NLLs on the test task. Note that the held-out tasks for some test tasks are the same because of the held-out rules in §E.2.1.

Comparing NLLs of the test tasks using models without training and trained via marginal likelihood like STBO, it is perhaps surprising to see that training on a subset of data points of the sub-dataset of the test task not only did not contribute to lowering NLL on the entire sub-dataset, but it even made it worse in 20 out of 23 test tasks. The training process by optimizing the NLL on a part of a sub-dataset leads to severe over-fitting. We can observe the same results of NLLs on all tasks. Without any training, our NLL is 148211.2. Yet single-task training leads to higher NLLs for all models trained on different sub-datasets.

Our method H*, on the other end, consistently achieves lower NLLs on both the test task and all tasks. Although it is not entirely clear what the relation is between a better NLL of the GP and better BO results, achieving lower NLLs typically means that the model has a better fit to the dataset. Hence, by the assumption of typical BO methods, the test function should look like a sample from our model, and so lower NLLs of model will contribute to matching the assumption of typical BO methods. By enhancing the assumption with ours on *i.i.d.* GP samples, Tab. 5 shows we then will be able to obtain models with a much better fit to the data.

Table 5: NLLs on 23 tasks and (pseudo) KL divergences on matching datasets with trained and randomly initialized GP models. The NLL of randomly initialized model (No training) on all tasks is 148211.2. The KL value of randomly initialized model (No training) is 2177.2. Training on a subset of a sub-dataset in the test task (Single task) often leads to much worse marginal likelihood on the entire sub-dataset. Training on irrelevant tasks (H*) achieves much lower (pseudo) KLs on matching datasets and lower NLLs for both the test task only and all tasks.

| | NLL of the test task only | | | NLL of all tasks | | (Pseudo) KL | |
|---|---|---|---|---|---|---|---|
| Test task | No training | Single task | H* | Single task | H* | Single task | H* |
| WMT XFormer 64 | −301.1 | 159.1 | **−1735.0** | 1147900.5 | **2264.5** | 9651.9 | **−40.2** |
| Uniref50 Transformer 128 | −651.7 | **−6829.4** | −1850.0 | 106348128.0 | **867.9** | 316672.2 | **−25.1** |
| LM1B Transformer 2048 | −540.6 | **−2009.7** | −1692.7 | 18840458.0 | **3565.7** | 57744.1 | **−23.5** |
| SVHN WRN 1024 | 9703.1 | 72407.5 | **4267.1** | 3399330.0 | **9346.5** | 4677.9 | **−0.9** |
| SVHN WRN 256 | 10770.0 | 53245.5 | **3794.8** | 1164804.5 | **9346.5** | 3092.7 | **−0.9** |
| ImageNet ResNet50 256 | 1196.7 | 7483.0 | **−746.3** | 7925583.5 | **−74.2** | 15028.1 | **−30.6** |
| ImageNet ResNet50 512 | 1300.2 | 6930.3 | **−673.1** | 1778823.5 | **−74.2** | 9462.1 | **−30.6** |
| MNIST CNNPoolTanh 2048 | 10079.7 | 38871.9 | **794.8** | 1375930.1 | **97.0** | 3165.5 | **−32.4** |
| MNIST CNNPoolTanh 256 | 12147.7 | 25607.9 | **550.0** | 556254.6 | **−606.0** | 1255.1 | **−41.9** |
| MNIST CNNPoolReLU 2048 | 26870.5 | 7149.3 | **5506.6** | 46538.2 | **1542.2** | 113.8 | **−59.4** |
| MNIST CNNPoolReLU 256 | 15601.6 | 6734.6 | **51.0** | 88687.7 | **−782.2** | 361.2 | **−41.5** |
| MNIST CNNReLU 2048 | 13939.2 | 40619.2 | **3153.2** | 743233.1 | **−231.4** | 877.6 | **−61.7** |
| MNIST CNNReLU 256 | 10111.0 | 34412.4 | **1365.3** | 977295.0 | **−779.8** | 1373.3 | **−46.2** |
| Fashion CNNPoolTanh 2048 | 2072.8 | 11433.0 | **−381.0** | 1139702.4 | **−1051.7** | 1910.5 | **−37.8** |
| Fashion CNNPoolTanh 256 | 2800.7 | 4115.6 | **−251.4** | 1278018.0 | **−1051.7** | 4208.3 | **−37.8** |
| Fashion CNNPoolReLU 2048 | 4677.4 | 725.2 | **−405.2** | 69173.3 | **−1051.7** | 205.1 | **−37.8** |
| Fashion CNNPoolReLU 256 | 3925.7 | 4254.4 | **−755.7** | 296739.1 | **−1051.7** | 1027.1 | **−37.8** |
| Fashion CNNReLU 2048 | 4667.3 | 6778.1 | **251.9** | 193488.4 | **−1051.7** | 597.0 | **−37.8** |
| Fashion CNNReLU 256 | 3295.1 | 29348.6 | **−235.1** | 1526829.2 | **−1051.7** | 3341.4 | **−37.8** |
| CIFAR100 WRN 2048 | 1271.5 | 15813.7 | **−467.4** | 3306556.5 | **312.3** | 25593.7 | **−19.2** |
| CIFAR100 WRN 256 | 1957.6 | 5950.8 | **−510.9** | 3468309.0 | **11.7** | 9288.4 | **−25.9** |
| CIFAR10 WRN 2048 | 5220.6 | 4917.6 | **832.9** | 334488.8 | **1127.1** | 1040.4 | **−14.8** |
| CIFAR10 WRN 256 | 7819.1 | 32995.8 | **463.4** | 895691.2 | **847.4** | 1946.0 | **−19.6** |

We also computed the (pseudo) KL divergence across all matching datasets in the last columns of Tab. 5. See Appendix C for a comprehensive analysis on pseudo KL divergence for degenerate multivariate Gaussians. Note that pseudo KL divergence can be negative. Here we use pseudo KL divergence if required by the matching dataset. Again, single-task training leads to unstable (pseudo) KL values, sometimes even higher than without training (2177.2). On the contrary, training with H* leads to much more stable and lower values for KL. This indicates that the model learned to predict

---

[8]All tasks include ImageNet ResNet50 2048. But it is excluded in the test tasks in Tab. 5 because it has much fewer data points than the others.

similarly to the sample mean/covariance estimate, which is known to help better selection of BO query points by Theorem 1.

**Batch evaluation.**    For simplicity of this paper, we did not consider batch evaluation but rather only focused on the prior selection dimension of the challenges in BO. However, it is straightforward to adopt any batch BO methods in conjunction with HyperBO to support obtaining observations in parallel. For example, we can directly use batch methods in Snoek et al. (2012), Kathuria et al. (2016), or Wang et al. (2017) to replace line 5 of Alg. 1.

**High-dimensional and large scale data.**    Similar to batch BO, our method can also be naturally combined with most high-dimensional and large scale BO methods to offer more capabilities. For these cases, typically a probabilistic model different from vanilla GPs may be adopted. In line 2 of Alg. 1, it is straightforward to adapt our method to optimize the cumulative marginal likelihood in Eq. 3 instead for the new model. Our meta-learning idea in this paper in fact also brings benefit to high-dimensional and large scale BO methods so that they can better identify their critical special structures, e.g. low-dimensional embedding Wang et al. (2016), cylindrical kernels Oh et al. (2018) or additive Mondrian kernels Wang et al. (2018a).

**Different search spaces.**    Roughly speaking, there could be two circumstances for difference search spaces. Case I is that tasks share the same search variables, but the search ranges for some variables are different. For example, we may have each function $f_i : \mathfrak{X}_i \rightarrow \mathbb{R}, i \in [N]$ and $\mathfrak{X}_i = \prod_{j=1}^{d}[l_{ij}, h_{ij}] \subset \mathbb{R}^d$. In this case, our solution still applies by simply setting a union search space as $\mathfrak{X} = \bigcup_{i=1}^{N} \mathfrak{X}_i$ for learning and use the designated search space of new tasks for optimization.

Case II is more complicated: the search space for each function $f_i$ is $\mathfrak{X}_i \subset \mathbb{R}^{d_i}$ and each dimension of $\mathfrak{X}_i$ may have a different meaning than another search space $\mathfrak{X}_j$ $(i \neq j)$. This paper does not have a solution for this scenario. Further research will be needed to reduce Case II to Case I which can be then immediately combined with HyperBO.

