# OpenReview forum: "Automatic prior selection for meta Bayesian optimization with a case study on tuning deep neural network optimizers"
_ICLR.cc/2022/Conference — ICLR 2022 Submitted_

### Official Review · Reviewer_7A1X · 2021-10-29

**Correctness:** 3
**Technical Novelty And Significance:** 2
**Empirical Novelty And Significance:** 1
**Recommendation:** 3
**Confidence:** 4

**Main Review:**

The problem of "warmstarting" HPO by making use of data from previous experiments is an obvious idea, and it has seen a large amount of past work, much of which the authors of this submission do not seem to be aware of, neither apparently was (Wang, 2018b) which seems more of a theoretical paper. In particular, there is quite a lot of work which uses GP models and scales linearly in terms of the number of past experiments, contrary to what is stated in the introduction. Two of the most interesting ones are maybe [1], [2]. The authors here cite (Perrone, 2018), which has these citations and more, so it is pretty odd the authors do not mention (or compare against) any of them.

Given the straightforward nature of what is proposed here (a setup closely related to what is done in [3]), I'd be quite surprised if for example [1] would not outperform it. After all, the assumption that data from experiments on quite different models can be modeled by the same mean and covariance function, is pretty strong. There are all sorts of issues with this idea, for example what if data from some tasks is much larger than data from others? Moreover, in what is proposed here, the surrogate model parameters do not even seem to be adapted to the current task, even as data from it becomes available. Here, methods like [1], [2] seem much more compelling to me, as they try to for example rank previous experiments by closeness to the current one. [1] is doing this without having to define any meta-features of the dataset, and also of course without relying on observations at the same configurations (given you model your data with a GP, you should certainly not need that anyway).

The experiments are not meaningful, because essentially all relevant prior work is missing for comparison. The authors more or less compare their proposal (in two variants) against a bunch of baselines, as if there was no revelant prior work. In fact, they even seem to invent on their own methods to compare against, such as "MIMO", in a way which has never been used for transfer HPO. Why? Please read about and compare against relevant prior work. Given they cite work (e.g., Perrone 2018), they should have been aware.

Apart from that, I also do not get much out of the experimental setup. Why was it chosen that way? Does it have any practical relevance? Does anybody else use this learning rate schedule, or was it just made up for this paper? I also did not find a discussion of a pretty critical point: how are the datapoints chosen for tasks you offline train on? In order to be realistic, these would have to be active choices themselves, because that is data we could have been obtained by running BO on them. Instead, my suspicion is that past data was sampled randomly, which would correspond to pure exploration (random search). Such data is obviously more valuable to obtain a good surrogate model fit, but also more expensive to obtain in the real world (one would have to run random search).

[1] Feurer etal: Practical Transfer Learning for BO, https://arxiv.org/abs/1802.02219
[2] Wistuba etal: Two-stage transfer..., ECML 2016
[3] Golovin etal: Google Vizier, KDD 2017

**Summary Of The Paper:**

This paper is concerned with speeding up Bayesian optimization by using evaluation data from previous, related tasks defined over the same configuration space. The authors propose to model the data from each experiment (or "task") by independent Gaussian processes, which all share the same mean and covariance function. This surrogate model can be learned from past data.

The paper also presents experiments on a fairly simple search space of 4 optimizer parameters. This is done for a bunch of datasets and NN models. And there is a pretty simple extension of theoretical results from (Wang, 2018b).

**Summary Of The Review:**

This paper proposes a simple idea for wam-starting BO by fitting the parameters of a GP surrogate model on past data. Unfortunately, a lot of relevant prior work is ignored here and not compared against. Instead, the proposed approach is compared against simple baselines, as well as methods that mostly seem to have been made up (such as "MIMO").

---

> ### Author Response · Authors · 2021-11-13
> **Response to Reviewer 7A1X (Part 1/2)**
>
> We thank reviewer 7A1X for the review. Please let us know if the following addresses your comments or if anything needs to be further clarified.
>
> We’d like to ask the reviewer to carefully re-read our assumption in Section 2. In our model, there is only one GP that generates i.i.d. samples of functions in different optimization tasks. There are no multiple “independent GPs”. In Section F, we did a thorough analysis on why the assumption is valid.
>
> About related work, there is indeed a vast amount of literature on warmstarting, end-to-end learning, meta learning and multitask BO etc. We focused on comparing our method to recent papers such as (Volpp et al., 2020) that was shown to outperform previous methods.
>
> For the specific works mentioned, [1] is not peer-reviewed and has 1 citation by (Perrone, 2018), which did not compare with [1]. As far as we know, [1] does not have open sourced code to compare with. We compared our method to (Volpp et al., 2020) in Section E.2.4, which showed much better performance than [2] and was the state-of-the-art we knew of with open sourced code. Although we intended to include more extensive comparisons to (Volpp et al., 2020), their models take very long to train due to policy learning based on simulations of BO loops (In our experiments, each training took more than 12 hours). Since other baselines in our paper already showed better and more stable performance than (Volpp et al., 2020), we did not perform further experiments with (Volpp et al., 2020). It’s also worth noting that experiments in (Volpp et al., 2020) assume knowledge of the optimal GP hyperparameters, but our method does not.
>
> One distinct advantage of our method over ensemble methods like [1] and (Golovin et al., 2017) is that once prior data is used to train the GP hyperparameters, we no longer need to keep the kernel matrices of prior data. In fact, we can perform SGD steps over data from an infinitely large set of prior tasks.
>
> Re concerns on baselines: We highly recommend reviewer 7A1X to take a look at the more detailed descriptions of baselines in Section E.1 and E.2.4. We carefully selected baselines that include competitive methods (e.g. STBOH, RFGP, MIMO), recent state-of-the-art (MAF) and methods widely used in practice (STBO). Deep ensemble and random feature GP are natural model choices for performing BO based on a large number of contextual observations.
>
>
> Q: “What if data from some tasks is much larger than data from others? ”
> A: Our case study is on deep learning optimization tasks and so all outputs share similar ranges. However, for the situation 7A1X mentioned, our method can potentially resolve it by adding an adaptive output warping; i.e. one output warper for each task so that one GP can fit the data nicely. Alternatively, a natural extension of our work is to use a mixture of GPs to replace a single GP. These are independent orthogonal directions to our current paper. We aim to explore these challenges as a future work.
>
> (Part 1/2)

---

> > ### Author Response · Authors · 2021-11-13
> > **Response to Reviewer 7A1X (Part 2/2)**
> >
> > Q: “The surrogate model parameters do not even seem to be adapted to the current task, even as data from it becomes available.”
> > A: This is exactly the value of this work. Our work challenges the status quo and current intuitions that ARD is a necessary element in BO. Our work shows that when one can leverage observations from related functions, ARD is not necessary (empirically and theoretically).
> >
> > However, as we mentioned in “Bayesian viewpoint” of Section 2, one can certainly choose to do the full Bayesian update. We did some preliminary comparisons and did not observe statistically significant differences between full Bayesian update and our current solution HyperBO. Given the much better runtime of HyperBO, we did not pursue a formal comparison to full Bayesian update.
> >
> > Q: “I also do not get much out of the experimental setup. Why was it chosen that way? Does it have any practical relevance? Does anybody else use this learning rate schedule, or was it just made up for this paper?”
> > A: As mentioned in Section 1 and further explained in Section 4.1, we collected this dataset because it has practical relevance to how the deep learning community does model training and hyperparameter tuning. Linear decay is a natural and common decay schedule. This learning rate schedule is simple and the specific linear decay schedule, or nearly identical ones, are commonly used in deep learning (e.g. in https://arxiv.org/abs/1811.03600 and https://arxiv.org/abs/1910.05446); it is implemented in open source code of (Gilmer et al., 2021). Anyone who opt to use the training code https://github.com/google/init2winit from (Gilmer et al., 2021) can make use of our method/data to tune the hyperparameters. Our paper is an example on how to set up HyperBO for existing machine learning models.
> >
> > Q: “I also did not find a discussion of a pretty critical point: how are the datapoints chosen for tasks you offline train on?”
> > A: We don’t restrict how data points are chosen in the proposed method HyperBO. In dataset PD1, we clearly described how our data points are selected in both matched and unmatched data: they are all uniformly-sampled hyperparameter points in the search space defined in Table 2. Random samples are much more efficient to obtain than running BO because different hyperparameters can be evaluated in parallel. If one needs to have BO generated data instead, it is easy to run offline BO experiments to re-order our randomly sampled data.
> >
> > (Part 2/2)

---

> ### Comment · Reviewer_7A1X · 2021-11-19
> **Reply to author feedback**
>
> I remain highly unconvinced about this work, and the author feedback does not really address my concerns. This works seems "to challenge the status quo" quite a bit, combining it with an extremely simple setup and a highly unconvincing empirical evaluation, against pretty esoteric methods.
>
> The authors did not reply to my request for sensible competitive baselines.
>
> I'd also like to point out that the authors confuse "Gaussian process" with "covariance function". The former is a random process of dependent variables (unless it is a noise process), the latter is a model for a random process. What they are doing is they share a covariance and mean function across different tasks, each of which is assigned a GP. This is one of the simplest baselines one could imagine, and if it was properly compared to competitive methods (like Feurer etal, or old work from Wistuba etal), it would likely fare not well. Instead, it is still not clear to me how the authors picked the methods to compare against, some of which (MIMO) are totally unrelated to transfer HPO.

---

> > ### Author Response · Authors · 2021-11-19
> > **We did reply to the request for competitive baselines**
> >
> > Hello, please take a look at the two comments we've already made as the response. Have you overlooked one by any chance? We replied point by point to all your concerns.
> >
> > Why do you think that we confuse "Gaussian process" with "covariance function"? We did not mention a single word about "covariance function" in the response. Again, as mentioned in our reply, we hope you could carefully re-read our assumption in Section 2. In our model, there is only one GP that generates i.i.d. samples of functions in different optimization tasks. There are no multiple “independent GPs” or one GP for each task.
> >
> > Regarding sensible competitive baselines, we already replied the following in the beginning of our response, where [1] and [2] are referring to your reference: "[1] Feurer etal: Practical Transfer Learning for BO, https://arxiv.org/abs/1802.02219 [2] Wistuba etal: Two-stage transfer..., ECML 2016".
> >
> > "About related work, there is indeed a vast amount of literature on warmstarting, end-to-end learning, meta learning and multitask BO etc. We focused on comparing our method to recent papers such as (Volpp et al., 2020) that was shown to outperform previous methods.
> >
> > For the specific works mentioned, [1] is not peer-reviewed and has 1 citation by (Perrone, 2018), which did not compare with [1]. As far as we know, [1] does not have open sourced code to compare with. We compared our method to (Volpp et al., 2020) in Section E.2.4, which showed much better performance than [2] and was the state-of-the-art we knew of with open sourced code. Although we intended to include more extensive comparisons to (Volpp et al., 2020), their models take very long to train due to policy learning based on simulations of BO loops (In our experiments, each training took more than 12 hours). Since other baselines in our paper already showed better and more stable performance than (Volpp et al., 2020), we did not perform further experiments with (Volpp et al., 2020). It’s also worth noting that experiments in (Volpp et al., 2020) assume knowledge of the optimal GP hyperparameters, but our method does not.
> >
> > One distinct advantage of our method over ensemble methods like [1] and (Golovin et al., 2017) is that once prior data is used to train the GP hyperparameters, we no longer need to keep the kernel matrices of prior data. In fact, we can perform SGD steps over data from an infinitely large set of prior tasks.
> >
> > Re concerns on baselines: We highly recommend reviewer 7A1X to take a look at the more detailed descriptions of baselines in Section E.1 and E.2.4. We carefully selected baselines that include competitive methods (e.g. STBOH, RFGP, MIMO), recent state-of-the-art (MAF) and methods widely used in practice (STBO). Deep ensemble and random feature GP are natural model choices for performing BO based on a large number of contextual observations."

---

### Official Review · Reviewer_jdBZ · 2021-10-31

**Correctness:** 3
**Technical Novelty And Significance:** 4
**Empirical Novelty And Significance:** 3
**Recommendation:** 8
**Confidence:** 4

**Main Review:**

### Reasons to Accept

+ It is well-written and well-organized.
+ It solves a very interesting problem, which transfers a history to the current task in Bayesian optimization setup.
+ Compared the work by Wang et al. (2018b), it solves more realistic setups.
+ It provides promising numerical results and sound theoretical results.

### Reasons to Reject

- I do not think that it degrades the contributions much, but four-dimensional search space is relatively small, compared to other Bayesian optimization or hyperparameter optimization papers.
- Following the above point, is there any specific reason why the authors use four-dimensional search space? I do not think this algorithm is not scalable. Moreover, for example, batch size can be one of the meta-parameters to be optimized.

### Questions to Authors

1. Can you elaborate why the proposed method does not train a GP model every iteration, e.g., every $t = 1, \ldots, T$? I think that it can be possible without (relatively) expensive computational costs.
1. H* NLL does not use a matching dataset, right? If you did not use multi-task GP regression, which has an additional input to indicate task information, does H* NLL (i.e., optimizing Equation (2) with $D_N$) work appropriately?

**Summary Of The Paper:**

This paper suggests a meta Bayesian optimization strategy that optimizes free parameters of GP including a prior function and noise variance, where multiple sets of historical observations are given. In particular, the proposed method chooses a free parameters using one of three approaches: (i) optimizing a marginal likelihood, (ii) measuring KL divergence, (iii) considering both marginal likelihood and KL divergence. The authors finally show the theoretical analyses on regret bounds and the numerical results on hyperparameter optimization.

**Summary Of The Review:**

I think that this paper addresses an interesting problem and suggests a novel method as described above. Thus, I would like to recommend acceptance.

---

> ### Author Response · Authors · 2021-11-13
> **Response to Reviewer jdBZ**
>
> We thank reviewer jdBZ for the review. Please let us know if the following addresses your comments or if anything needs to be further clarified.
>
> Q: “Is there any specific reason why the authors use four-dimensional search space? I do not think this algorithm is not scalable.”
> A: Agreed, it's true that there's no reason why our method wouldn't scale to much larger dimensions.  We wanted to motivate our method by choosing a set of hyperparameters that generalize across many, many DL workloads regardless of architecture, modality, etc. so we chose optimizer parameters (which ended up being relatively low-D).
> As detailed in “High-dimensional and large scale data” of Section F, our method focuses on the GP model and training objective which is perpendicular to aspects such as scalability to input dimension and data size. So it is true that our method combining other existing approaches can potentially also scale to much higher input dimensions or data size.
>
> Q: “batch size can be one of the meta-parameters to be optimized.”
> A: We indeed considered it in the beginning but batch size is closely related to the memory size of the machine we run training jobs on. Ideally, one would set the batch size as large as possible as long as a batch of data points fit the memory constraints. Because we do not assume machines can be chosen, we also do not assume batch size can be chosen. Hence the batch size is more relevant to the experiment configurations or task descriptions rather than a hyperparameter.
>
> Q1: “Can you elaborate why the proposed method does not train a GP model every iteration,...?”
> A: We choose not to train a GP model every iteration to save computational cost. In the experiments, our TrainGP (line 2 of Alg 1) typically takes a few hours to complete. Hence we fix the GP parameters and only do posterior inference on new data in a given BO task.
>
> Q2: “H* NLL does not use a matching dataset, right? If you did not use multi-task GP regression, which has an additional input to indicate task information, does H* NLL (i.e., optimizing Equation (2) with $D_N$) work appropriately?”
> A: For H* NLL, the matching dataset and the non-matching one are combined across tasks to compute NLLs. In our experiments, we have one dataset $D_N$ which contains both matching and non-matching data. For H* KL, we extract the matching dataset in D_N to compute the KL divergence in Eq. (4).
> If I understand correctly, the second part of this question is asking if there is a single task, whether H* NLL works appropriately. If there is only one task, Equation (2) will become the type II data likelihood in a typical GP setup, a.k.a. Equation (2) without the sum. And that is exactly the situation where HyperBO becomes a typical BO. However, if the second part of this question is asking whether H* NLL would work if there is no task information, the answer would be no, neither H* NLL nor H* KL would work if we are given data without knowing which data points belong to which task. Our assumptions on having a multi-task dataset would be invalid without any task information.

---

> > ### Comment · Reviewer_jdBZ · 2021-11-24
> > **Response to Authors**
> >
> > Thank you for your thoughtful comment.
> >
> > Most of my questions and concerns are resolved.
> >
> > However, in particular, a relatively small search space is a weak point of this work, while I do not agree with that point.
> >
> > I think four or five hyperparameters of interest are a good design choice.
> >
> > Beyond such a search space, I cannot imagine which search space in hyperparameter optimization is appropriate for this meta-Bayesian optimization scenarios, and which search space in neural architecture search is the most effective for finding an optimal architecture.
> >
> > But, if you find some realistic and large space as a target search space, this paper would be better.
> >
> > Best regards,
> >
> > Reviewer jdBZ.

---

### Official Review · Reviewer_dwbG · 2021-11-02

**Correctness:** 2
**Technical Novelty And Significance:** 1
**Empirical Novelty And Significance:** 1
**Recommendation:** 3
**Confidence:** 4

**Main Review:**

The reviewer appreciates the authors putting effort into the empirical evaluation of the proposed method. However, the proposed approach is not interesting to the Bayesian optimization community and is trivial to some degree. The reviewer believes that the targeting problem presented in this work is a very important one and an effective method could be of great practical value.

In the abstract, authors claim that "data from similar functions" could lead to a better prior for GP. Obviously a better prior for GP is desirable and that is why the marginal likelihood is used to optimize  parameters of a GP. From such a claim, it is expected that an efficient method for BO will be presented by exploring novel similarities between tasks. However, throughout this paper, there is no definition of a similarity between tasks and tasks are treated as independent. This raises my concern on this work's novelty, which is my biggest concern on this paper.

Authors claim that the critical difference between this work and standard BO algorithms is the initial learning process in line 2 of Algorithm 1. The corresponding likelihood of this approach is given in eq(2). I do not get the point how this approach is different from existing GP modeling and eq(2) is simply the unnormalized marginal likelihood for all data points since all tasks are assumed to be independent. Such a formulation is not only trivial to the GP community , but also to the empirical Bayes community.


Additional (minor) issues:
1. the graphical model for GP in Figure 1 is wrong
2. there exist a lot of inconsistencies in this paper. In assumptions section, it is assumed the variance is known however the variance is a hyper-parameter in the marginal likelihood.
3. lots of claims and statements are superfluous. For example, authors claim one limitation of existing approaches is the total number of BO iterations must be set in a manual way. However, throughout this paper, the number of iterations is still pre-defined. What is the point of saying this is a limitation while not touching it at all? Another example, authors claim  "interpretability of intermediate steps" is lost in existing methods, however, this problem is not touched either.
4. Another contribution of this paper is a tuning dataset. I can see the value of such a dataset, however, failing to explicitly describe the required computation resources makes claiming this being a contribution less convincing.

**Summary Of The Paper:**

This paper presents a Bayesian optimization method based on meta-BO. The motivation is tasks can share the same parameter structure and this shared information, e.g. correlation between tasks, can be transferred to new and similar tasks. An example is to optimize the the hyper-parameters of a same optimizer across different architectures and different datasets. This problem is a very important one in the community of Bayesian optimization and a reasonable method can lead to a potentially dramatic decrease in the required computation, especially when the objective function is very expensive. This work tries to overcome limitations of existing methods. For example, the method proposed in this work does not need to evaluate all objective functions associated with all tasks on the same parameters.

**Summary Of The Review:**

The proposed method is trivial. The theoretical part presented in this paper is very minimal and incremental.

---

> ### Author Response · Authors · 2021-11-13
> **Response to Reviewer dwbG (Part 1/2)**
>
> We thank reviewer dwbG for the review. Please let us know if the following addresses your comments or if anything needs to be further clarified.
>
> Q: “there is no definition of a similarity between tasks and tasks are treated as independent”
> A: The reviewer raised a point that is exactly the key insight our paper conveys: similar functions could be represented by independent functions sampled from the same distribution. Because of that, we specifically used a relatively vague word similarity instead of being close in a certain distance metric. In “Bayesian viewpoint” of Section 2, we further explained how our independent assumption on functions is in fact “conditionally independent” (conditioned on the GP) under a hierarchical Bayes interpretation. Thus, without the conditioning on the GP, all functions are still correlated because they are all samples of the same GP distribution.
> It is indeed natural for readers to raise these concerns because our paper is operating in a regime where frequentist methods meet Bayesian paradoxes. We highly recommend that reviewer dwbG take a closer look at “Bayesian viewpoint” of Section 2 to ease the confusion.
>
> Q: “how this approach is different from existing GP modeling and eq(2) is simply the unnormalized marginal likelihood for all data points since all tasks are assumed to be independent.”
> A: We understand the confusion. First, eq(2) is NOT the unnormalized marginal likelihood for all data points. Although functions are conditionally independent, all data points observed on each function are highly correlated. Please point out to us a reference where the exact assumptions and Eq. (2) are derived. We will cite that if it exists. In fact, the simplicity of Eq. (2) is what we pursue. As mentioned in the response to jdBZ, if there is only one task, Equation (2) will become the type II data likelihood in a typical GP setup, a.k.a. Equation (2) without the sum.
>
> Q: “the graphical model for GP in Figure 1 is wrong”
> A: Please point to us where it is wrong.
>
> Q: “there exist a lot of inconsistencies in this paper. In assumptions section, it is assumed the variance is known however the variance is a hyper-parameter in the marginal likelihood.”
> A: If noise variance is what the reviewer is referring to, we do not assume the noise variance is known. In Section 2, we specifically mentioned “Notice that we do not assume that the mean function µ, kernel k and noise variance $sigma^2$ are given.” If the reviewer is referring to “... perturbed by i.i.d. additive Gaussian noise with known variance…”, we meant additive Gaussian noise given a fixed variance parameter. We understand this is confusing and updated our manuscript changing to “... perturbed by i.i.d. additive Gaussian noise…”.
>
> Q: “authors claim one limitation of existing approaches is the total number of BO iterations must be set in a manual way.”
> A: We do not need to pre-define the number of BO steps (T) for a new task. The “For loop” in line 4 can be replaced by “While True”. We believe this is clear since T is not an input to TrainGP. The rest of the algorithm is BO itself and also does not require a fixed T. However, in (Chen et al., 2017; Volpp et al., 2020), the number of BO steps has to be determined before training by design. That is to say, model training itself in (Chen et al., 2017; Volpp et al., 2020) requires a prefixed number of BO steps (T) that will be performed on any new tasks.
>
> Q: “authors claim "interpretability of intermediate steps" is lost in existing methods, however, this problem is not touched either”.
> A: The full sentence is “by nature of using a highly parameterized model to train the strategy, we lose the interpretability of intermediate steps that GPs and acquisition functions provide.” We believe it is clear that we are referring to the interpretability of GPs and acquisition functions, which are both preserved components in our method. Our method is structured in the same way as regular BO and we do not use a highly parameterized model. On the contrary, end-to-end learning approaches such as (Chen et al., 2017; Volpp et al., 2020) predict the point to evaluate with complex highly parameterized neural-networks-based models, which loses the nice interpretability of GPs and acquisition functions in regular BO.
>
> (Part 1/2)

---

> > ### Author Response · Authors · 2021-11-13
> > **Response to Reviewer dwbG (Part 2/2)**
> >
> > Q: “I can see the value of such a dataset, however, failing to explicitly describe the required computation resources makes claiming this being a contribution less convincing.”
> > A: All details about how to run the same models/datasets/optimizers are made clear in Justin M. Gilmer, George E. Dahl, and Zachary Nado. init2winit: a jax codebase for initialization, optimization, and tuning research, 2021. URL http://github.com/google/init2winit. We generated our tuning dataset with this open sourced codebase. Anyone can run these models on any machine with enough memory to fit a batch of data points. Moreover, given the tuning dataset, there is no need to run our data collection experiments to generate more data. A researcher can simply take the dataset and do transfer learning/meta learning with HyperBO to optimize their own model/dataset of interest. We used a total of roughly 12,000 machine-days of compute to run the approximately 50,000 hyperparameter evaluations (aka model training with different hyperparameters) in our dataset. And that is 12,000 machine-days saved for anyone who would like to use or analyze the dataset.
> >
> > (Part 2/2)

---

### Official Review · Reviewer_Neri · 2021-11-03

**Correctness:** 4
**Technical Novelty And Significance:** 3
**Empirical Novelty And Significance:** 2
**Recommendation:** 5
**Confidence:** 3

**Main Review:**

I have a few key concerns about this paper.

- Why fixed hyperparameters? This is clearly the bottleneck of Metalearned BO, and if these hyperparameters are learned offline, this seems to (A) somewhat eliminate the strength of HyperBO which is the linear scaling per task ---obviously this still helps significantly during the offline training, but still a point of concern of mine, and (B) seems not robust, especially if the set of representative completed tasks is heavily biased.

- HyperBO, in the experiments, uses the PI acquisition function. Is there a particular reason why this is? PI is quite greedy (even more than EI), so is there any intuition as to why PI is appropriate in this situation.

- In Figure 2b, I am somewhat concerned about the empirical performance of HyperBO. Though it beats the baselines, it does so in a 4D search space, using thousands of tasks; this seems like overkill. The error bars are also all over the place. This is somewhat unfair of me to ask for I admit, but I am curious if a much simpler approach involving restricting the search space (given that it is fixed) will help (see the paper “Learning search spaces for Bayesian optimization, Perrone et al., 2019). I feel like there is definitely enough data for this to make a difference.

- Also, the experiments only really concern one optimization problem involving optimizer hyperparameters. Though this one experiment is quite impressive in terms of the data involved, iit would be nice to see another experiment (say for tasks that might be easier like tuning a random forest).


**Summary Of The Paper:**

 HyperBO assumes the tasks are independent given the hyperparameters, unlike typical metalearning approaches which assume tasks are related. This allows for an efficient Kronecker decomposition of the kernel and thus linear, rather than cubic scaling, across tasks.

Using this model, HyperBO performs BO as usual; maximize the acquisition function to obtain the next point to evaluate. HyperBO also makes the critical assumption of an offline pre-training of hyperparameters on a representative set of completed tasks; during optimization itself the hyperparameters are fixed.


**Summary Of The Review:**

I have some concerns about the assumptions used in the methodology, as well as the experiments, which leave a number of open questions. In particular, the fixing of GP hypers seems to largely remove the need for scaling, which is the primary strength of HyperBO. Furthermore, though the experimental set up uses a large amount of data to achieve somewhat unconvincing results in my mind, and only one optimization problem is presented (though worth noting, is thoroughly analyzed). Thus, I can't recommend acceptance at the time.

---

> ### Author Response · Authors · 2021-11-18
> **Response to Reviewer Neri**
>
> We thank reviewer Neri for the review. Please let us know if the following addresses your comments or if anything needs to be further clarified.
>
> Q: “This allows for an efficient Kronecker decomposition of the kernel and thus linear, rather than cubic scaling, across tasks”
>
> A: What you are describing is the multi-task Gaussian Process that was explored by Swersky et al. 2013, which uses the multi-task kernel of Bonilla et al., 2007. This technique has a requirement that we must evaluate different functions (tasks) at the same set of inputs. One of the key contributions of this work is to remove the requirement of having the same inputs across tasks.
>
> If we write down the full kernel matrix for our proposed method (which we don’t do explicitly; it would be an inefficient way of implementing our algorithm), it is a block diagonal matrix where each block corresponds to a task. Note that this block diagonal kernel matrix CANNOT be decomposed into a Kronecker product.
>
> Despite the connections to multitask GP, we want to again highlight the fact that we are not just using a simplified multitask GP: we model different functions as independent samples from the same GP; multitask GP, in contrast, attempts to model the correlations among the functions.
>
> Reference:
> Edwin V Bonilla, Chris Williams, Kian M Chai. "Multi-task Gaussian process prediction." Advances in neural information processing systems (2007): 153-160.
>
> Q: “Why fixed hyperparameters?”
>
> A: See the joint reply.
>
> Q: “...PI acquisition function. Is there a particular reason why this is? PI is quite greedy (even more than EI), so is there any intuition as to why PI is appropriate in this situation.”
>
> A: Please find the reason in the first paragraph of Section 4.2. More comparisons of different acquisition functions including EI can be found at Section E.5 (Figure 13). All HyperBO variants still outperform the best alternatives.
>
> Q: “4D search space, using thousands of tasks; this seems like overkill.”
>
> A: We’ll address the 4D search space in a joint reply. Note that we do NOT have “thousands of tasks”. As shown in Table 1, we at most have 23 training tasks. In fact, Figure 2 (more descriptions in E.2.1) only uses at most 18 irrelevant training tasks. The number of training tasks in Figure 3 varies from 3 to 23.
>
> Q: “The error bars are also all over the place.”
>
> A: As explained in the caption, the shaded areas are not error bars. They are 20/80 percentiles. Table 3 has the actual standard errors and it shows the statistical significance of how our method is better than the most competitive baselines.
>
> Q: “a much simpler approach involving restricting the search space (given that it is fixed) will help (see the paper Learning search spaces for Bayesian optimization, Perrone et al., 2019)”.
>
> A: Learning search spaces is orthogonal to learning the GP prior. Ideally one can combine both methods to put BO in practice. Also it does not seem true that (Perrone et al., 2019) has a simpler approach than ours. HyperBO naturally eliminates the “bad” regions through the learned mean function. On the contrary, (Perrone et al., 2019) relies on constrained optimization to make use of the learned search space.
>
> Q: “the experiments only really concern one optimization problem involving optimizer hyperparameters.”
>
> A: See joint reply for “4D search space is not convincing; only experiments for NN optimizers.”

---

### Author Response · Authors · 2021-11-18
**Answers to common questions of some reviewers**

Q: 4D search space is not convincing; only experiments for NN optimizers.

A: We wanted to motivate our method by choosing a set of hyperparameters that generalize across many, many DL workloads regardless of architecture, modality, etc. so we chose optimizer parameters (which ended up being relatively low-D).

In fact, as shown in Table 2 of the winning submission of NeurIPS 2020 Black-Box Optimisation Challenge (https://arxiv.org/pdf/2012.03826.pdf), the tasks typically have low-dimensional hyperparameters, on the same order as ours. We believe the NN optimizer hyperparameter tuning problem is very representative of how BO is used in practice.

Indeed there are a lot of BO papers that verify their methods by experimenting on a selection of black-box function optimization scenarios. Our paper is different because we are building on an existing method (Wang et al., 2018) and making it more practical by case-studying a very realistic and difficult task: optimizing NN optimization hyperparameters. Our work can be viewed as a more practical and general version of the ideas that exist in (Wang et al., 2018) and (Perrone, 2018). (Wang et al., 2018) and (Perrone, 2018) already showed that this idea works across a variety of cases: synthetic functions, grasp optimization, pose and placement optimization, SVM hyperparameter optimization etc. However, these tasks are relatively simple and synthetic compared to our NN optimizer tuning tasks based on http://github.com/google/init2winit. Our training datasets consist of popular benchmarks including large scale datasets such as LM1B and ImageNet, while our models are selected from state-of-the-art deep learning literature.

Generalize to higher dimensions: As detailed in “High-dimensional and large scale data” of Section F, our method focuses on the GP model and training objective which is perpendicular to aspects such as scalability to input dimension and data size. So it is true that our method combining other existing approaches can potentially also scale to much higher input dimensions or data size.


Q: Fixing hyperparameters seems to be a weakness.

A: Fixing hyperparameters is exactly the value of this work. Our work challenges the status quo and current intuitions that ARD is a necessary element in BO. Our work shows that when one can leverage observations from related functions, ARD is not necessary (empirically and theoretically).

However, as we mentioned in “Bayesian viewpoint” of Section 2, one can certainly choose to do the full Bayesian update. We did some preliminary comparisons and did not observe statistically significant differences between full Bayesian update and our current solution HyperBO. Given the much better runtime of HyperBO, we did not pursue a formal comparison to full Bayesian update.

---

### Decision · Program_Chairs · 2022-01-20

**Decision:**

Reject

**Comment:**

This paper claims a practical improvement over one of earlier meta BO methods. Warm-starting BO or HPO by making use of data from past experiments or tasks seems to be interesting and useful for some applications. In fact, there are a large amount of work on this topic, but a lot of relevant prior work is ignored in this paper unfortunately. I appreciate the authors for making efforts in responding to reviewers’ comments. However, after the discussion period, most of reviewers had serious concerns in this work, pointing out that the proposed method is rather trivial and the comparison is made only against a simple baseline. It was also suggested to improve the experiments. While the idea is interesting, the paper is not ready for publication at the current stage.